


# Portable Calibrator for NO Based on Photolysis of $N_2O$ and a Combined $NO_2/NO/O_3$ Source for Field Calibrations of Air Pollution Monitors

John W. Birks, Andrew A. Turnipseed, Peter C. Andersen, Craig J. Williford, Stanley Strunk, Brian Carpenter and Christine A. Ennis

2B Technologies, 2100 Central Ave., Boulder, CO 80301 USA

*Correspondence to:* John Birks (johnb@twobtech.com)

**Abstract.** A highly portable calibration source of nitric oxide (NO) based on photolysis of nitrous oxide ($N_2O$) supplied by 8- or 16-g disposable cartridges is demonstrated to serve as an accurate and reliable transfer standard for the calibration of NO monitors in the field. The instrument provides output mixing ratios in the range 0-1,000 ppb with a precision and accuracy of better than the greater of 3 ppb or 3 % of

the target NO mixing ratio over a wide range of environmental conditions of ambient temperature (8.5-35.0 °C), pressure (745-1,015 mbar corresponding to 2.7-0.0 km elevation) and relative humidity (0-100 % RH). Combination of the NO calibration source with a previously described ozone calibration source based on photolysis of oxygen in air provides a new instrument capable of outputting calibrated mixing ratios of NO, ozone ($O_3$) and nitrogen dioxide ($NO_2$), where the $NO_2$ is produced by the stoichiometric gas-phase reaction

of NO with $O_3$. The portable $NO_2/NO/O_3$ calibration source requires no external gas cylinders and can be used for calibrations of NO, $NO_2$ and $O_3$ instruments for mixing ratios up to 1,000, 500, and 1,000 ppb, respectively. This portable calibrator may serve as a convenient transfer standard for field calibrations of ozone and $NO_x$ air pollution monitors.

## 1. Introduction

Measurements of gaseous nitric oxide (NO), nitrogen dioxide ($NO_2$) and ozone ($O_3$) are critical in numerous fields. NO is a direct combustion product that is readily oxidized in air to form $NO_2$. Ground-level ozone is produced by the photochemical interactions between $NO_x$ (NO + $NO_2$) and organic

compounds in sunlight (Haagen-Smit, 1957). Both $NO_2$ and $O_3$ are known to produce several cardiac and respiratory ailments (both acute and chronic) and are classified as "criteria pollutants," and their atmospheric levels are regulated by the U.S. Environmental Protection Agency (U.S.-EPA) and corresponding regulatory agencies around the world. Verification of compliance with these regulations requires a comprehensive and continuous monitoring system of both ambient atmospheric levels as well as



NO$_x$ emissions from industrial combustion sources (often referred to as CEMS – Continuous Emissions Monitoring). Routine NO$_x$ monitoring is also required for safety reasons in settings where diesel engines and machinery are used in confined areas, such as in the mining industry. Nitric oxide is used in the medical field where it is approved by the U.S. Food and Drug Administration for treatment of adult pulmonary arterial hypertension (Abman, 2013) and for persistent pulmonary hypertension in hypoxic term and near-

term newborns ("blue babies") (Clark et al., 2000). During this inhalation therapy, the concentration of NO (typically 20 parts per million) must be continuously monitored during administration, and, very importantly, the concentration of its toxic impurity NO$_2$ must also be continuously monitored and kept below 3 parts-per-million by volume or ppm (preferably much lower due to the acute NO$_2$ toxicity).

          In all these applications, accurate monitoring of NO, NO$_2$ and O$_3$ not only requires stable, robust

chemical analyzers, but also a way to test the validity of the analyzer response periodically using a standardized calibration method. Ideally, this is done by introducing gaseous standards with well-known concentrations of the analyte of interest. The frequency of calibration depends upon the species being measured and the instrumental approach. A detailed discussion of ozone detection methods and calibration protocols is given in a previous publication describing a portable ozone calibrator (Birks et al., 2018b) and

will not be repeated here. In the past, NO$_x$ calibration methods were developed primarily for use with analyzers based on the chemiluminescence (CL) reaction of NO with an excess of ozone, which is the most widely used method for quantifying NO and, following its conversion to NO, NO$_2$ (Fontijn et al., 1970; Ridley and Howlett, 1974; Kley and McFarland, 1980; Steffenson and Stedman, 1974; Demerjian, 2000). These analyzers require relatively frequent calibrations to assess both the basic instrumental sensitivity drift

for NO as well as the NO$_2$ conversion efficiency.

          Calibration of monitors for NO is typically achieved by use of gas standards. A well-known problem with NO gas standards is that NO is unstable in gas cylinders at low concentrations; when NO standards are prepared at low part-per-billion by volume (ppb) levels, there is a strong tendency for the concentration of NO in the cylinder to decline with time even though the NO is diluted into an unreactive

gas such as nitrogen. This is because NO is thermodynamically unstable with respect to disproportionation to form N$_2$O and NO$_2$ according to the equilibrium (Burkholder et al., 2015):

$$3\ NO \rightleftharpoons N_2O + NO_2 \qquad \Delta H^o_{298} = \text{-157.6 kJ/mol} \qquad (1)$$

Although extremely slow in the gas phase, this reaction may be catalyzed on the interior walls of compressed gas cylinders. The walls may be treated to slow the reaction, but the treatment is not always

effective, and one cannot be certain that the concentration of NO in a gas cylinder is what it was when the cylinder was last analyzed. Furthermore, even trace amounts of oxygen (O$_2$) in the diluent gas can react to oxidize NO to NO$_2$ according to the reaction (Atkinson et al., 2004):





$$2\,NO + O_2 \rightarrow\ 2\,NO_2 \quad k_2 = 2 \times 10^{-38}\ cm^6\ molec^{-2}\ s^{-1}\ at\ 298\ K \tag{2}$$

Because of reaction 2, compressed gas standards for NO cannot be made with air as the diluent. This is a disadvantage since it is desirable to calibrate NO instruments using the same diluent gas as the gas being analyzed, which most commonly is air. Nitric oxide standards are much more stable at high concentrations; thus, it is common to prepare gas standards at a high ppm level in an unreactive diluent gas such as $N_2$ and then dynamically dilute that standard with air prior to entering the analytical instrument being calibrated. Even at high ppm levels, NIST-certified NO gas standards are typically only certified for 1-2 years. Although the dynamic dilution method works quite well, it is difficult to use as a portable transfer standard due to the need for a cylinder of certified NO gas mixture and the need for accurately calibrated flow meters, whose response can vary with temperature.

Nitrogen dioxide gas standards in standard passivated aluminum cylinders are known to degrade over a relatively short period of time regardless of concentration (U.S.-EPA, 2019). The development of a $NO_2$ primary reference standard and subsequent calibration traceability protocols is an ongoing project (U.S.-EPA, 2019). Historically, the U.S.-EPA has recommended two methods for dynamic multipoint calibration of $NO_2$ analyzers based on chemiluminescence (Ellis, 1975; U.S.-EPA, 1983): one based on a permeation tube source of $NO_2$ and another based on the gas phase titration (GPT) technique. Although the permeation tube source has found acceptance in certain areas (e.g., mine safety, Chilton et al., 2005), the difficulty of producing stable and reproducible $NO_2$ outputs from permeation tubes has precluded them from widespread use. The GPT technique is almost exclusively used for calibrating analyzers for compliance with the U.S. Clean Air Act. In the GPT method, the instrument is first calibrated for NO by the dynamic dilution of a high concentration $NO/N_2$ gas standard traceable to a NIST Standard Reference Material (SRM) with $NO_x$-free air. The instrument is then calibrated for $NO_2$ by addition of varying concentrations of ozone to an excess of NO. The 1:1 stoichiometric conversion of NO to $NO_2$ via the reaction of NO with $O_3$ (Burkholder et al., 2015),

$$NO + O_3 \rightarrow\ NO_2 + O_2 \qquad k_3 = 1.9 \times 10^{-14}\ cm^3\ molec^{-1}\ s^{-1}\ at\ 298\ K \tag{3},$$

forms the basis of the calibration. Ozone concentrations are generated by photolysis of $O_2$ (typically from air) and added to an excess of NO while allowing for sufficient mixing time so that reaction 3 goes to completion. Nitrogen dioxide is calibrated based on the increase in $NO_2$ signal ($NO_x$ – NO in CL analyzers) relative to the decrease in the NO signal (U.S.-EPA 2002). $NO_2$ formed should equal the NO consumed if the $NO_2$ conversion efficiency to NO of the analyzer to be calibrated is unity. Incomplete conversion yields $[NO_2]_{formed} < [NO]_{consumed}$, such that using the GPT reaction as a calibration incorporates a measure of the conversion efficiency for analyzers where NO is monitored (i.e., CL analyzers). However, as with the case



above concerning NO, a portable means of $NO_2$ calibration via the GPT method requires a NIST-SRM NO gas mixture, a source of purified air, some type of ozone generator, and accurate mass flow controllers.

More recently, several new techniques that directly measure $NO_2$ based on variations of UV absorption (e.g., cavity ringdown and cavity-attenuated phase-shift spectroscopy) have become available (Paldus and Kachanov, 2006; Kebabian et al., 2005; Kebabian et al., 2008). However, many of these do

not measure NO. Therefore, for $NO_2$-only analyzers the GPT calibration method requires either (1) a second instrument that can measure the loss of NO or (2) a NIST-traceable ozone source, such that the loss of ozone can be correlated with the formation of $NO_2$. Note that the standard GPT calibration procedures can still be applied to methods that directly measure $NO_2$ and then indirectly measure NO (the opposite of the chemiluminescence technique) - such as in the long-path folded tubular photometer (FTP) developed in our

group that measures direct $NO_2$ absorbance at 405 nm (Birks et al., 2018a).

In this paper we will initially describe and evaluate a portable calibration source for nitric oxide based on the photolysis of $N_2O$ (the 2B Technologies Model 408 Nitric Oxide Calibration Source™; Andersen et al., 2019) and show that it is suitable to be used as a field transfer calibration standard. An advantage of this approach to NO calibration is that the nitrous oxide can be supplied by an 8- or 16-g

cartridge (e.g., whipped cream chargers), thereby eliminating the need for a compressed gas cylinder. The result is a highly portable NO calibrator. Recently, we have combined this Model 408 NO Calibration Source with a Model 306 $O_3$ Calibration Source™ (described in Birks et al., 2018b) to produce a GPT $NO_2$ calibrator (the Model 714 $NO_2$/NO/$O_3$ Calibration Source™). Here we evaluate the feasibility of using this instrument as a portable transfer standard for $NO_2$ without the requirements of having a certified gas

standard and accurately calibrated mass flow controllers, thus increasing the portability of the transfer standard. Finally, we show that combining the $O_3$ calibrator and the NO calibrator into one instrument enables the user to perform robust calibrations for all 3 gases ($NO_2$, NO and $O_3$) using just one highly portable instrument suitable for laboratory or field applications.

## 2. Theory of operation

### 2.1 Nitric oxide (NO) calibration


The Model 408 Nitric Oxide Calibration Source™ (2B Technologies, Boulder, Colorado) makes use of a low-pressure mercury (Hg) lamp to photolyze pure nitrous oxide ($N_2O$) to produce NO. The vacuum UV emission lines of mercury near 184.9 nm are absorbed by $N_2O$ to produce electronically excited oxygen atoms, O ($^1$D),

$$N_2O + h\nu \rightarrow N_2 + O(^1D) \tag{4}$$





where hv symbolizes a photon of light. These highly energetic oxygen atoms react with $N_2O$ with a near collisional reaction rate coefficient ($k = 1.3 \times 10^{-10}$ cm$^3$ molec$^{-1}$ s$^{-1}$) to form three different sets of products

$$O(^1D) + N_2O \quad \rightarrow \quad 2\ NO \qquad \phi = 0.61 \pm 0.03 \qquad (5a)$$
$$\rightarrow \quad N_2 + O_2 \qquad \phi = 0.39 \pm 0.03 \qquad (5b)$$
$$\rightarrow \quad N_2O + O(^3P) \quad \phi < 0.01 \qquad (5c)$$

with the branching ratios ($\phi$) shown (Burkholder et al., 2015). Since reaction 5a produces 2 NO molecules, the overall quantum yield for NO production is approximately 1.22. The NO calibration source is similar in design to our ozone calibrator (Birks et al., 2018b), as both make use of the 184.9-nm line of a low-pressure mercury lamp. An important fundamental difference is that the 184.9-nm absorption cross section for $N_2O$ is approximately 14 times larger than that of $O_2$. The absorption cross section, $\sigma$, of $N_2O$ at the

184.9-nm mercury emission line is ~$1.4 \times 10^{-19}$ cm$^2$ molec$^{-1}$ (Creasey et al., 2000; Cantrell et al., 1997) compared to ~ $1 \times 10^{-20}$ cm$^2$ molec$^{-1}$ for $O_2$ (Yoshino et al., 1992, Creasey et al., 2000). Also, $O_2$ is only 21 % of the air that passes through the photolysis chamber in the ozone calibrator, while $N_2O$ is supplied to the photolysis chamber by a source that is > 99 % $N_2O$. The result is that the $N_2O$ gas absorbs the 185-

nm light ~70 times stronger than does $O_2$ in air. At 298 K and 1 atm, the molecular concentration, c, is 2.46 $\times 10^{19}$ molec cm$^{-3}$; thus, the absorption of 184.9-nm light from the low-pressure mercury lamp becomes optically thick (1/e attenuation) at a path length, 1/($\sigma$c), of 0.3 cm, and 99 % of the light is absorbed for a path length of 1.35 cm. Under such conditions, the rate of production of NO (molecules cm$^{-3}$ s$^{-1}$) depends almost entirely on the lamp intensity and is independent of flow rate (i.e., residence time in the photolysis

cell). The NO/$N_2O$ stream exiting the photolysis chamber is diluted into $NO_x$-scrubbed air to produce the desired output concentration of NO in air. At constant flow rates of $N_2O$ and the dilution air, the concentration of NO in the calibrator's output is varied by merely changing the lamp intensity.

We typically observe a small amount of $NO_2$ produced from the NO photolytic generator ($\leq$ 3 % of the NO produced). This is likely due to the formation of O($^3$P) atoms in the photolysis cell, which

combine with NO via the reaction:

$$O(^3P) + NO + M \rightarrow NO_2 + M \qquad (6)$$

(where M is a third body, most likely $N_2O$ in this case). O($^3$P) atoms can arise from several possible sources. Nishida et al. (2004) report a quantum yield for O($^3$P) from $N_2O$ photolysis of $0.005 \pm 0.002$ (i.e., $NO_2/NO$ = 0.005/1.22 ~ 0.4 %). Quenching of O($^1$D) to O($^3$P) by $N_2O$ (reaction 5c) likely contributes up to another

0.8 % (Vranckx et al., 2008 report a limit of $\phi_{5c} < 0.01$ at 298K). Oxygen ($O_2$), which is a typical $N_2O$ impurity, can also photolyze to produce two O($^3$P) atoms. Even NO itself could be photolyzed at 184.9 nm ($\sigma$ ~ $3 \times 10^{-18}$ cm$^2$ molec$^{-1}$, thermodynamic dissociation threshold of 189.7 nm; Iida et al., 1986 and





Burkholder et al., 2015) to produce $O(^3P)$ atoms; however, this would be expected to be of lesser importance due to the relatively lower NO concentrations (ppm) within the photolysis cell. In all of these cases, the amount of $NO_2$ formed relative to NO should be small and approximately constant over time.

### 2.2 Ozone calibration

Our photolytic ozone calibration source has been described in detail previously (Birks et al., 2018b) and the following is only meant to briefly highlight the important points of this calibrator since it plays a key role in the $NO_2$ calibration device described in the following section. In the photolytic ozone calibration source that is used in our Model 306 Ozone Calibration Source™ (2B Technologies, Boulder, Colorado), a low-pressure mercury lamp produces calibrated concentrations of ozone by photolysis of oxygen in air:

$$O_2 + h\nu \rightarrow 2\,O \qquad (7)$$

$$(O + O_2 + M \rightarrow O_3) \times 2 \qquad (8)$$

$$\text{Net: } 3\,O_2 + h\nu \rightarrow 2\,O_3$$

A key difference compared to the NO photolytic source described above (Section 2.1) is that for a 1-cm path length (and at 1 atm and 25 °C), the $O_2$ absorption in air is nearly optically thin (~ 5 % light absorbed). For an optically thin system, the mixing ratio of ozone produced depends linearly on the residence time within the photolysis chamber; thus it varies with volumetric flow rate. In the Model 306 Ozone Calibration Source, the mass flow rate, temperature and pressure are continuously measured to compute the volumetric flow rate (and, therefore, the residence time), and the lamp intensity is adjusted in a feedback loop to maintain a constant ozone output mixing ratio. A further key point in the ozone calibration source is that the photolysis cell must be maintained at a constant (slightly heated) temperature to ensure constant overlap between the Hg lamp emission lines and the $O_2$ absorption lines and to maintain a constant ratio of lamp intensities between the 184.9 nm Hg line and the 253.7 nm Hg line that is monitored in the feedback loop to maintain a constant photolysis rate (Birks et al., 2018b). The main difference between the previously described Model 306 Ozone Calibration Source and the one used for the $NO_2$ calibrator described below (Section 2.3) is that the flow rate through the photolysis cell is much lower (~ 50 cm³ min⁻¹ as opposed to 3000 cm³ min⁻¹). This leads to longer residence times and higher ozone concentrations within the photolysis cell before subsequent dilution downstream. The minor repercussions associated with this modification are discussed in Section 4.

### 2.3 $NO_2$ calibration in a combined calibrator

Combining the above two calibrators into a single unit (the Model 714 $NO_2/NO/O_3$ Calibration Source™, 2B Technologies, Boulder, Colorado) makes it possible to calibrate not only for NO and $O_3$, but also a third gas, $NO_2$. Calibrated concentrations of NO and $O_3$ are produced as described above. Calibration





of $NO_2$ is accomplished via the gas phase titration (GPT) technique (reaction 3), making use of the NO and $O_3$ produced in the combined calibration source. Here, $O_3$ is reacted with an excess of NO to produce known concentrations of $NO_2$, under conditions such that $[NO]_{consumed} = [NO_2]_{produced}$. A key difference in the Model 714 from the two individual calibrators is that reaction (3) must be carried out at high concentrations (ppm level) to drive reaction (3) to completion; therefore the NO and $O_3$ reagents are mixed

before subsequent dilution. Modeling the second-order kinetics of reaction (3) (see Fig. 1) shows that with $[NO] = 2 \times [O_3]$ (i.e., NO a factor of 2 in excess of $O_3$), ~5 ppm of NO is required to consume 99.6 % of the ozone for a reaction time of 4 seconds. Increasing the reaction time allows for lower [NO] to be used to obtain the same completeness of reaction. It should be noted that for $NO_x$ analyzers that measure both NO and $NO_2$, it is not necessary for reaction (3) to go to completion because one measures the consumption

of NO relative to the production of $NO_2$. However, residual ozone complicates the calibration protocol as reaction (3) continues to alter the $NO/NO_2$ ratio (at a reduced rate after dilution) as the gas mixture is transported to the analyzer to be calibrated. Thus, the $NO/NO_2$ ratio would depend on the residence time of the connection tubing. For $NO_2$-only analyzers, it is critical to have > 99 % conversion as the $NO_2$ signal produced may be correlated to the loss of ozone (which is assumed to be at the calibrated target

concentration and typically not explicitly measured). The calculations described in Fig. 1 can be used as a guide to the required concentrations and residence times of a GPT reactor.

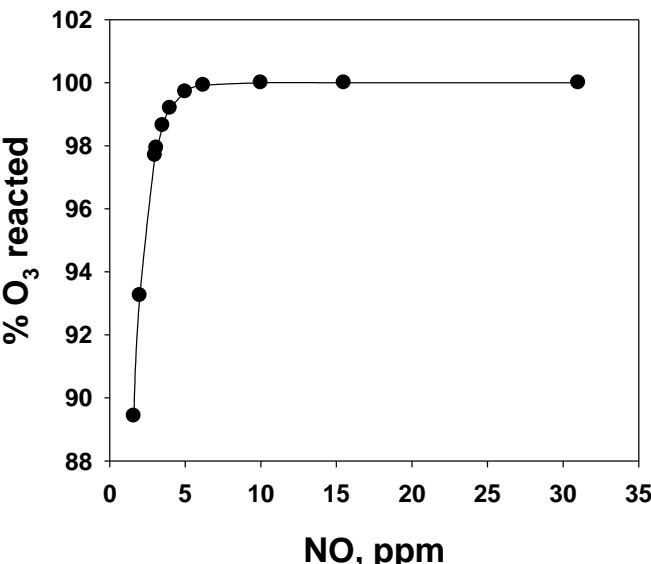

**Figure 1.** A plot of the percent of $O_3$ consumed vs. initial NO concentration for the conditions of $[NO] = 2 \times [O_3]$ and a total reaction time of 4 seconds.



One more point concerning the GPT chemistry is that NO must be maintained in excess over ozone (Ellis, 1975; U.S.-EPA, 2002). If ozone is used in excess, $NO_2$ can react with the excess ozone to produce

$NO_3$,

$$NO_2 + O_3 \rightarrow NO_3 + O_2 \tag{9}$$

and $NO_3$ can subsequently form $N_2O_5$ rapidly via (Bertram et al., 2012):

$$NO_2 + NO_3 \rightleftharpoons N_2O_5 \tag{10f, 10r}$$

Reaction (9) is ~600 times slower than reaction (3) ($k_9 = 3.22 \times 10^{-17}$ cm$^3$ molec$^{-1}$ s$^{-1}$ at 298 K, Burkholder

et al., 2015), but can proceed to a small extent at ppm levels of $NO_2$ and $O_3$. At room temperature and $NO_2$ concentrations greater than about 25 ppb, reaction (10) favors $N_2O_5$ formation and proceeds relatively rapidly ($k_{10}$ ~ $1.4 \times 10^{-12}$ cm$^3$ molec$^{-2}$ s$^{-1}$, Burkholder et al., 2015), thus resulting in a net loss of 2 $NO_2$ molecules. In typical CL analyzers that use heated molybdenum to convert $NO_2$ to NO, $N_2O_5$ production is not observable, since the heated catalyst will thermally decompose $N_2O_5$ rapidly (reaction 10r), followed

by reduction of both $NO_2$ and $NO_3$ to NO – thus, not affecting the observed $[NO]_{consumed} = [NO_2]_{produced}$. However, in the case of photolytic NO converters and the long-path FTP method mentioned in the previous section, the formation of $N_2O_5$ would cause an underestimate in the calibration (i.e., $[NO]_{consumed} >$ $[NO_2]_{produced}$). For photolytic converters, there would be no way to elucidate the error as the lower observed $NO_2$ would likely be incorporated into an incorrect conversion efficiency.

**3.    Experimental**

***3.1    Portable Nitric Oxide Calibration Source***

Figure 2 is a schematic diagram of the NO calibration source. An air pump draws ambient air into the instrument through $NO_x$ and ozone scrubbers to produce the diluent air stream. The air flow rate is measured by a mass flow meter and is controlled by use of restrictors (not shown) and a needle valve that

vents part of the flow. The needle valve is adjusted to produce diluent air at a total output volumetric flow rate of ~3 L min$^{-1}$. In the most portable configuration of the instrument, nitrous oxide is supplied by a cartridge containing either 8 or 16 grams of liquid $N_2O$ with a headspace pressure of ~50 atmospheres at 20 °C. A combined cracker/regulator punctures the cartridge as it is tightened and also drops the outlet delivery pressure to below 25 psig. A 25 psig pressure relief valve is installed inside the instrument housing

to prevent over pressurization. The valve on the cracker/regulator provides a coarse adjustment of the $N_2O$ flow rate. A voltage sensitive orifice (VSO) valve is then used to provide fine control of the $N_2O$ flow rate to $60 \pm 1$ cm$^3$ min$^{-1}$ in a feedback loop. Pressure within the gas stream is measured but not controlled. The $N_2O$ then passes through an aluminum photolysis cell (volume ~ 6 cm$^3$) where a small fraction of the $N_2O$



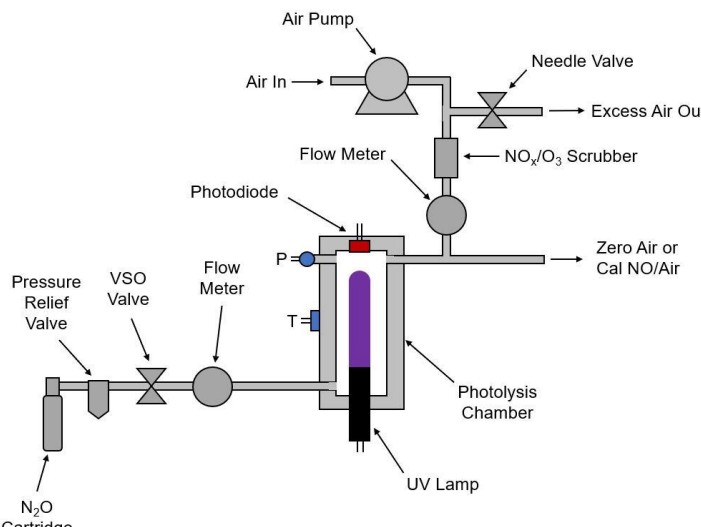

**Figure 2.** Schematic diagram of the 2B Technologies Model 408 Nitric Oxide Calibration Source.

is converted to NO (and $N_2 + O_2$) by a low-pressure mercury discharge lamp. As discussed above, because

the system is optically thick (essentially every photon is absorbed), the NO production rate (molecules s$^{-1}$) is independent of photolysis cell pressure and $N_2O$ flow rate, the production rate depending only on the lamp intensity. Since nearly all of the 184.9 nm light is absorbed by the $N_2O$ in the cell, the lamp intensity at the 253.7 nm mercury line is monitored by a photodiode and controlled by the microprocessor. As the monitoring wavelength (253.7 nm) and the photolysis wavelength (184.9 nm) are different, it is important

to maintain a constant ratio of lamp emission at these two wavelengths. This is accomplished by regulating the photolysis cell (which houses the lamp) at a constant temperature of 40 °C. The NO/$N_2O$ stream exiting the photolysis chamber is diluted into the ~3 L min$^{-1}$ flow of NO$_x$-scrubbed air to produce the desired output concentration of NO in air. The photodiode voltage (i.e., a measurement of lamp intensity) is calibrated against the output NO concentration as measured by a NO analyzer that has been recently calibrated using

a NIST-SRM NO gas standard/dilution system. Note that for the NO calibration source to be a valid transfer standard, the photolytic NO source must be validated against a NIST-traceable NO standard to provide a lamp intensity vs. NO output concentration working curve.

Nitrous oxide can be supplied to the instrument either by means of $N_2O$ cartridges (commercially available and often used as whipped-cream chargers) as shown in Fig. 2, or by connection to a lecture bottle

or gas cylinder containing $N_2O$. The cartridge holder and cracker allow use of either 8- or 16-g cartridges containing liquid $N_2O$ and will supply a gas flow of $N_2O$ of 60 cm$^3$ min$^{-1}$ for approximately 1.2 or 2.5 hours,


respectively. Alternatively, a lecture bottle or tank of $N_2O$ may be used, allowing continuous operation of 1.5 days for a lecture bottle containing 227 g or 174 days for a QA cylinder containing 27 kg of $N_2O$.

### 3.2 Combined $NO_2$, NO and $O_3$ Calibration Source

A portable calibrated source of $NO_2$ can be achieved by combining the NO photolytic calibration source (described in Section 2.1 and 3.1) with the photolytic ozone calibration source (described in Section 2.2 and in Birks et al., 2018b). This is commercially available as the 2B Technologies Model 714 $NO_2/NO/O_3$ Calibration Source™, which is capable of providing calibrated mixing ratios of $NO_2$, NO or $O_3$. A schematic diagram of the Model 714 $NO_2/NO/O_3$ Calibration Source is shown in Fig. 3. The

instrument produces $O_3$ by photolysis of oxygen in air, NO by the photolysis of $N_2O$, and $NO_2$ by gas phase titration (GPT) of known concentrations of $O_3$ in an excess of NO.

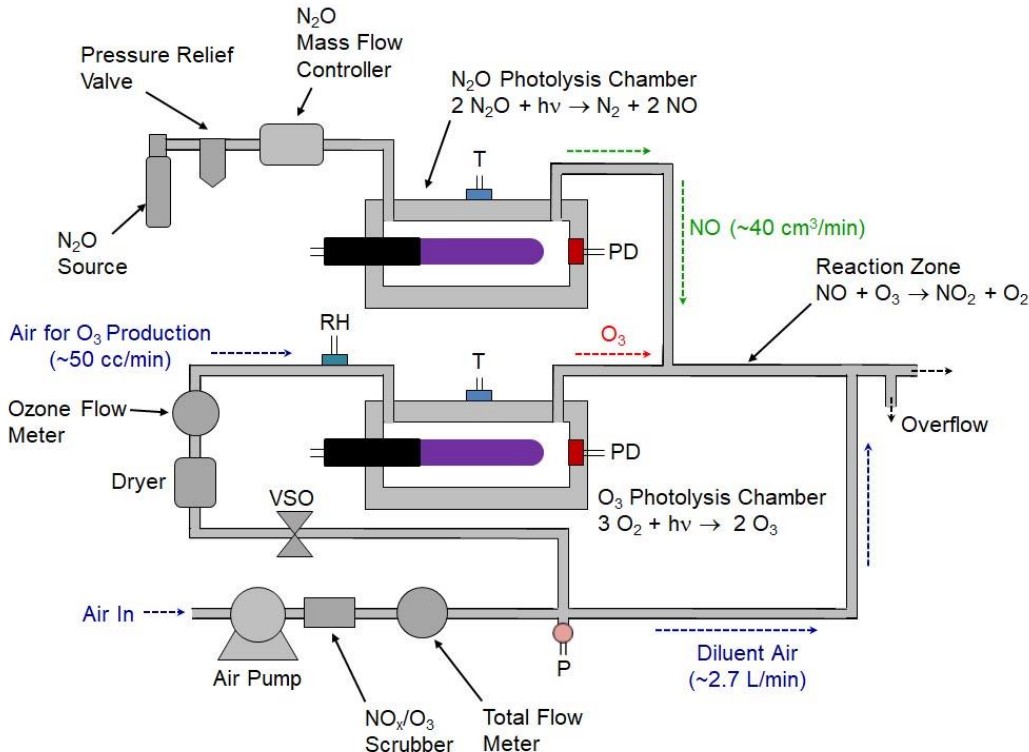

**Figure 3.** Schematic diagram of the 2B Technologies Model 714 $NO_2/NO/O_3$ Calibration Source.





An air pump pushes ambient air through an $O_3$/$NO_x$ scrubber; thereafter the flow is split using a restrictor to send a volumetric flow of ~50 cm$^3$ min$^{-1}$ through an ozone photolysis chamber, with the bulk of the flow, ~2.7 L min$^{-1}$, serving as a diluent gas. This main flow combines with the effluent of the photolysis cells just prior to the outlet. A voltage sensitive orifice (VSO) valve controls the flow, as measured by a mass flow meter, through the photolysis chamber. For production of NO, a pressurized source of $N_2O$ passes through a mass flow controller and into a NO photolysis chamber at a volumetric flow rate of ~40 cm$^3$ min$^{-1}$. Flows through the $O_3$ and NO photolysis chambers join at a tee prior to entering a reaction zone having a volume of 6.5 cm$^3$ consisting of 20.3 cm of 6.4-mm i.d. Teflon tubing. When the instrument is in "$NO_2$" mode (making $O_3$ and an excess of NO), the $O_3$ is quantitatively converted to $NO_2$ during the ~4.3-s residence time. The high concentration $O_3$, NO or NO/$NO_2$ mixture is diluted by a factor of ~30-54 (the larger being in the absence of the $N_2O$ flow for outputting only $O_3$) with $O_3$/$NO_2$-scrubbed ambient air at a tee just prior to the instrument outlet. Ozone or NO is produced at calibrated concentrations by turning the corresponding lamps on and adjusting their intensities as measured by photodiode measurements in the respective chambers. Typically, the $N_2O$ flow is turned off when only ozone is being output to conserve $N_2O$ usage. To produce known mixing ratios of $NO_2$, calibrated amounts of $O_3$ (corrected for the slight dilution by $N_2O$) are produced in the range 0-500 ppb with the NO output set at least twice the output ozone (e.g, 1,000 ppb of NO is required for 500 ppb of $O_3$ to be converted to $NO_2$). Note that these are the concentrations exiting the calibrator as opposed to the much higher concentrations found within the reaction zone.

### 3.3 Validation as a suitable transfer standard

As described in Section 1, the U.S.-EPA sets out procedural guidelines for calibration of monitors used for regulatory monitoring of ambient $O_3$ and $NO_2$ (U.S.-EPA, 2002; U.S.-EPA, 2013). Since most $NO_2$ monitors actually monitor NO (CL analyzers), their guidelines also describe calibration of NO as a matter of necessity even though NO is not a criteria pollutant. For $NO_x$, the basis of these procedures ties NO and $NO_2$ calibrations to a NIST-traceable SRM gas mixture of NO. However, the U.S.-EPA does not provide guidance for transfer standards that do not include the direct use of a either a NIST-SRM gas mixture or a gas mixture that is somehow traceable to a NIST-SRM (as with the photolytic $NO_2$ calibrator described here). In contrast, specific statistical requirements are established for the use of either photolytic generators or analyzers based on photometry for use as transfer standards in the calibration of ozone monitors (U.S.-EPA, 2013; Birks et al., 2018b). In lieu of direct statistical requirements for a photolytic NO and $NO_2$ transfer standard, we have applied the same requirements that are established for a Level 4 ozone transfer standard (U.S.-EPA, 2013). Level 4 ozone transfer standards must undergo a "6 × 6" verification in which six calibration curves, each consisting of six approximately equally spaced



concentrations in a range including 0 and 90 % (± 5 %) of the upper range of the reference standard, are obtained on six different days (U.S.-EPA, 2013). The relative standard deviations of the six slopes of the calibration plots must not exceed 3.7 %, and the standard deviation of the 6 intercepts cannot exceed 1.5 ppb.

        The "6 × 6" verification requires an analyzer whose calibration is traceable to a NIST standard.
For the measurements presented here, we have used either a 2B Technologies Model 205 or 202 ozone monitor as a reference photometer for $O_3$, and a 2B Technologies Model 400 (for NO only) or a Model 405 (NO and $NO_2$) as reference analyzers for $NO_x$. The ozone monitors (Model 202 or 205) are each certified as an ozone Federal Equivalent Method (FEM) by the U.S.-EPA (EQOA-0410-190) and are NIST-traceable through comparison with our currently accredited ozone calibrator (Thermo Scientific Model 49i-PS). The
Model 405 also is designated as an FEM for $NO_2$ (EQNA-0217-243), and it and the Model 400 were calibrated using a Teledyne-API Model 700 Dynamic Dilution Calibrator using a NIST-traceable NO gas mixture (Scott Specialty or Airgas). Furthermore, the ozone photometer (used to measure the ozone for the GPT reaction) within the Model 700 was also calibrated against our NIST-traceable Thermo Scientific ozone standard.

Because a goal of the calibration sources is their use in field calibrations of analyzers, we also need to consider the effect of environmental factors such as temperature, pressure and humidity on the output mixing ratios of the photolytically generated analytes. The factors of optical opacity and the photochemistry discussed above in Section 2 imply that environmental variables such as temperature, pressure and relative humidity should have minimal effects on the performance of the ozone and NO
calibrators. However, this assumes that the $N_2O$ is completely optically thick and that the sensors for temperature, pressure and mass flow rate are perfectly linear and independent of one another. For example, as we point out below, mass flow meters depend on molecular composition and will not be perfectly accurate when the water vapor mixing ratio changes. We have previously described similar tests of the ozone calibrator (Birks et al., 2018b), and here we carry out additional environmental tests of the NO
calibration source. Furthermore, temperature certainly affects the GPT chemistry (reaction 3) and can place limitations on the usable concentration ranges – typically at low concentrations where reaction (3) may not go to full completion. The methods for varying the temperature, pressure and humidity will be described as the results are presented in the following section.






## 4.    Results and discussion

### 4.1    NO Calibration Source

The Model 408 Nitric Oxide Calibration Source was first introduced as a product by 2B
Technologies in 2007 but has not been described in the scientific literature. Applications of this highly
portable NO calibrator have been limited primarily because users need an instrument that also calibrates
for $NO_2$ measurements. The photolytic NO generator described here has since been used in the 2B
Technologies Model 211 Scrubberless Ozone Monitor, where NO serves as a gas-phase scrubber, and, more
recently, the Model 714 $NO_2/NO/O_3$ Calibration Source. The NO generator is identical in all three
instruments – the only differences being the $N_2O$ flow rates used and the degree of dilution. Since, as we
discussed in Section 2.1, 1 atm of $N_2O$ is optically thick at 184.9 nm, the flow rate of $N_2O$ through the
chamber is not critical. Because essentially every photon is absorbed, the production rate of NO is
determined only by the lamp intensity – a low flow rate of $N_2O$ through the chamber produces the same
number of molecules of NO per second as a high flow rate. The effect of changing the $N_2O$ flow rate is
only to change the total flow into which the NO produced is diluted, which is small since the $N_2O$ flow rate
is only 1-2 % of the total flow. Optical opacity was verified experimentally by using the NO calibration
source in a Model 714 and varying the $N_2O$ flow rate through the photolysis chamber from 5 to 50 $cm^3$
$min^{-1}$ and setting the lamp intensity to output a constant 400 ppb of NO. The resulting NO mixing ratios
measured are given in Table 1. As is readily apparent, there is no observable dependence of NO
concentration produced on the flow rate of $N_2O$ within the measured uncertainties. Due to this invariance

**Table 1**. NO mixing ratios measured from a Model 714 $NO_2/NO/O_3$ Calibration Source as a function of
the $N_2O$ flow rate.

| $N_2O$ flow rate $cm^3$ $min^{-1}$ | NO mixing ratio[a] ppb ($\pm 1\sigma$) |
|---|---|
| 5.1 | 394.5 ± 6.1 |
| 7.7 | 396.9 ± 1.4 |
| 10.1 | 397.9 ± 1.1 |
| 15.3 | 397.8 ± 3.9 |
| 20.4 | 398.4 ± 3.0 |
| 25.4 | 399.2 ± 4.1 |
| 30.4 | 397.1 ± 2.4 |
| 35.5 | 397.9 ± 1.9 |
| 39.6 | 394.7 ± 2.7 |
| 45.6 | 396.4 ± 2.6 |
| 47.7 | 396.2 ± 1.8 |

[a]NO measured using a Model 405 $NO/NO_2/NO_x$ monitor.





with $N_2O$ flow, this flow rate is often set based on the balance between $N_2O$ usage and the response time to
        a change in NO concentration. Faster flow rates result in a quicker flush time of the photolysis cell and
        lead to more rapid changes in NO concentration. For example, in the 2B Technologies Model 211
        Scrubberless Ozone monitor, only a constant amount of excess NO is required, and thus a small flow rate
        (10-15 $cm^3$ $min^{-1}$) serves to conserve $N_2O$ usage. However, in both the NO calibrators (the Model 408 and
714), higher flow rates (40-60 $cm^3$ $min^{-1}$) are used to allow for more rapid concentration changes (< 1
        minute).

        The NO calibration source is typically configured to deliver a calibration gas at a volumetric flow
        rate of 2.5 to 3.0 L $min^{-1}$. A change in flow rate of diluent air would be expected to change the concentration
        of NO produced. However, the instrument continuously measures the total mass flow rate and adjusts the
lamp intensity to compensate for changes in dilution so as to produce a constant output mixing ratio of NO.
        In typical operation, these intensity adjustments are small as the total flow rate is usually rather constant
        (within ± 5 %). However, the intensity vs. total flow rate feedback loop was tested by measuring the NO
        output (at 200 ppb) as the total flow rate was varied between 2.2 and 4.5 Lpm. There was no measurable
        difference in the NO mixing ratios (< ± 2 % or 4 ppb, data not shown).


#### 4.1.1   Precision, accuracy, stability and reproducibility of the photolytic NO Calibration Source

        An example of the NO output concentration of a Model 408 NO Calibration Source as measured
        with a recently calibrated 2B Technologies Model 400 Nitric Oxide Monitor™ (2B Technologies, Boulder,
        Colorado; Birks and Bollinger, 2006) over 2.5 hours is shown in Fig. 4. The NO calibrator was
programmed to run through a series of 10 target concentration steps of 0, 50, 100, 150, 200, 250, 200, 150,
        100 and 0 ppb with a hold time of 15 minutes at each concentration. As can be seen the rise (or fall) time
        between steps is on the order of 1 to 1.5 minutes (samples are every 10 sec) before stable NO outputs are
        established. Precisions (1σ) averaged 2.6 ppb at non-zero target concentrations and were not significantly
        different from the 2.9-ppb average of precisions of the first and last steps with the lamp off. This implies
that the observed variability was almost entirely due to the NO monitor used – thus NO output
        concentrations from the photolytic calibrator are stable to considerably better than ± 2.6 ppb. All output
        concentrations agreed with the target concentrations within one standard deviation, with the exception of
        the 250 ppb level where the measured output concentration was higher by 7.1 ± 2.7 ppb or 2.8 %.

        Figure 5 shows the temporal behavior and stability of NO produced from the NO calibration source
over the entire usable time period of an 8-gram $N_2O$ cartridge (~ 160 min) for a setpoint of 800 ppb. The
        small amount of $NO_2$ produced ($NO_2$/NO of 1.6 %) is also shown (note the break in the y-axis). Both NO
        and $NO_2$ were measured using a 2B Technologies Model 405 $NO_2$/NO/$NO_x$ Monitor. There is a slight



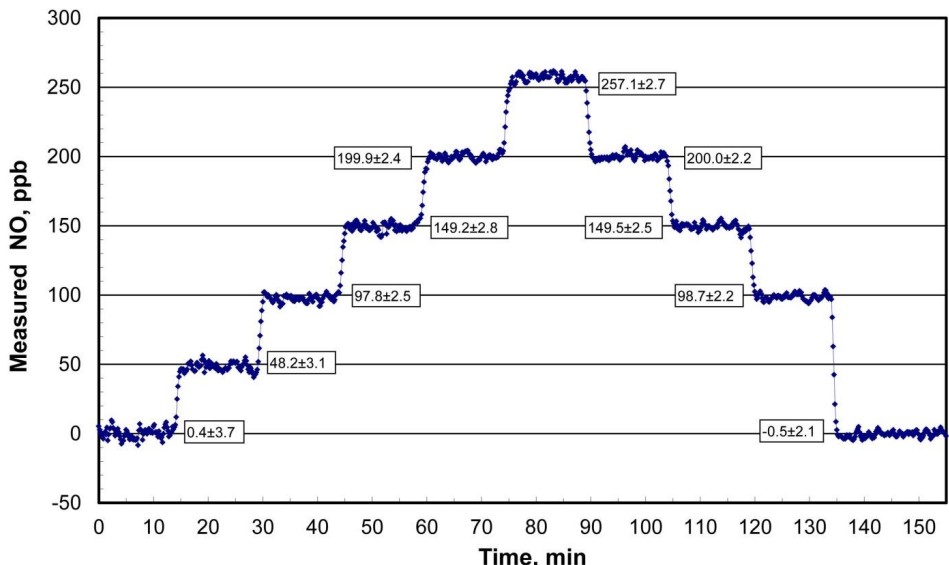

**Figure 4.** Measured output from Model 408 NO Calibration Source for programmed NO mixing ratios of 0, 50, 100, 150, 200, 250, 200, 150, 100, and 0 ppb and time steps of 15 minutes. Average concentrations after the 1-min step changes are shown along with standard deviations.

increase in the measured NO (4.4 ppb/hr) with a total NO increase of ~ 12 ppb (1.5 % of the 800 ppb setpoint) over the lifetime of the $N_2O$ cartridge. A similar experiment at a setpoint of 200 ppb (data not

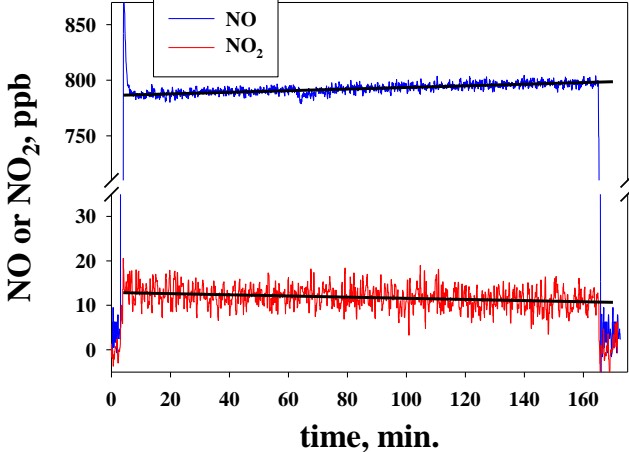

**Figure 5.** NO and $NO_2$ output by the NO Calibration Source over the time to use an entire 8-gram $N_2O$ cartridge. The NO setpoint was 800 ppb. Lines drawn are linear fits to data between 15 and 165 minutes.


shown) gave a similar percentage increase (3.8 ppb or +1.9 %). However, when using a 27 kg $N_2O$ cylinder with similar stated purity (99.5 %), no increase in NO was measured (< 0.3 % at a setpoint of 800 ppb) over the same 2.7-hour time frame. This suggests that the small 1-2 % increase in the NO signal may arise from preferential volatilization of the small amount of impurities in the $N_2O$ (likely $N_2$ and $O_2$), leading to a slightly more purified $N_2O$ over the lifetime of the 8-gram cartridge. This would be expected to be very

slow and unobservable when using a larger cylinder. Overall this suggests that the NO calibration source is stable to about 2 % over the 2- to 3-hour time span needed for conducting calibrations regardless of the $N_2O$ source.

        $NO_2$ showed a corresponding decrease of -0.8 ppb/hr (total of 2.2 ppb) over the course of depleting the 8-gram $N_2O$ cartridge (see Fig. 5). This small decrease is within the measurement precision of our $NO_2$

analyzer. No decrease in $NO_2$ could be detected at lower NO setpoints (e.g., 200 ppb) or when using a larger cylinder. Therefore, the $NO_2$ produced at a given NO setpoint is essentially constant over a several hours and would have minimal effect (< 1 %) on $NO_2$ calibrations described in Section 4.2.

        Using the NO calibration source from a Model 714 $NO_2$/NO/$O_3$ calibrator (S/N = 1014), a "6 × 6"
verification was undertaken to ascertain whether it could be used as a traceable transfer standard for NO.

Figure 6 shows the calibration plots obtained over 6 days using a recently calibrated 2B Technologies Model 405 $NO_2$/NO/$NO_x$ Monitor to detect the generated NO. Due to the high reproducibility, the results are also given in tabular form (Table 2), including the measured slopes, intercepts and correlation

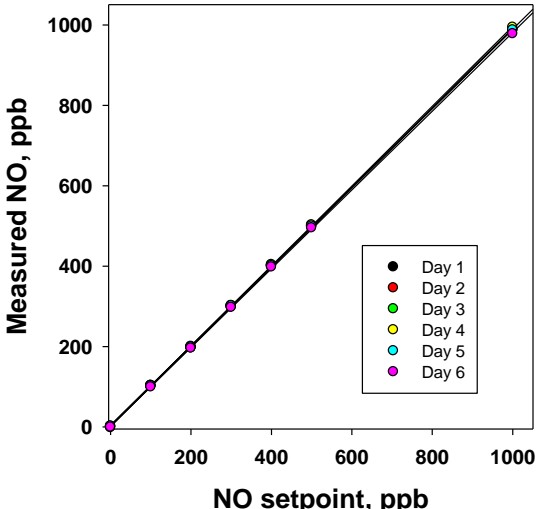

**Figure 6.** Plots of the NO measured (ppb) from the NO photolysis source of a 2B Technologies Model 714 $NO_2$/NO/$O_3$ Calibration Source vs. the NO setpoint on 6 consecutive days.

**Table 2**. Results from the "6 × 6" validation over six consecutive days for the NO photolysis source in the 2B Technologies Model 714 NO$_2$/NO/O$_3$ Calibration Source.

| NO Setpoint (ppb) | NO measured: Day 1 | Day 2 | Day 3 | Day 4 | Day 5 | Day 6 | Average (ppb) | Std. Dev. (ppb) |
|---|---|---|---|---|---|---|---|---|
| 1000 | 988.9 | 986.4 | 988.1 | 994.2 | 987.4 | 978.1 | 987.2 | 5.2 |
| 500 | 498.1 | 498.2 | 501.1 | 502.6 | 499.6 | 495.2 | 499.1 | 2.6 |
| 400 | 400.2 | 400.2 | 402.4 | 403.8 | 401.2 | 397.8 | 400.9 | 2.1 |
| 300 | 299.0 | 299.2 | 301.6 | 302.0 | 301.2 | 297.3 | 300.1 | 1.8 |
| 200 | 197.0 | 197.1 | 199.0 | 200.1 | 199.1 | 196.1 | 198.1 | 1.6 |
| 100 | 101.7 | 101.3 | 103.0 | 103.0 | 102.7 | 99.6 | 101.9 | 1.3 |
| 0 | -1.5 | -0.1 | 1.0 | 1.5 | 2.1 | -0.6 | 0.4 | 1.4 |
| | | | | | | | | |
| **Intercept:** | 1.3 | 2.2 | 4.0 | 3.6 | 4.2 | 2.2 | 2.9 | 1.2 |
| **Slope:** | 0.9897 | 0.9866 | 0.9872 | 0.9930 | 0.9856 | 0.9791 | 0.9869 | 0.004 |
| **R$^2$:** | 0.9999 | 0.9999 | 0.9999 | 0.9999 | 0.9999 | 0.9999 | 0.9999 | |

coefficients from a linear regression. As can be seen, day-to-day variations are not statistically different from the precision of the measuring analyzer (~ ± 2 ppb) with the exception of the highest (1000 ppb) point, which has a slightly higher standard deviation (± 5.2 ppb). However, this is still a precision that is < 1 % of the measured value. From the linear regressions it can be seen that the standard deviation in the intercepts is 1.2 ppb, below the 1.5 ppb required of Level 4 ozone transfer standards. Also the standard deviation in the slopes is only ± 0.004 or 0.4 %, which is substantially below the required level of 3.7 %. Therefore, it is obvious that the photolytic NO calibrator is highly stable and reproducible and successfully meets the same criteria set forth for the establishment of an ozone transfer standard.

### 4.1.2 Effects of temperature, pressure and humidity on the photolytic NO Calibration Source

In order to test for the effect of temperature on the NO concentrations produced, we made measurements of the output mixing ratio of a Model 408 NO Calibration Source, using a program consisting of steps of 0, 50, 100, 150, and 200 ppb at ambient temperatures of 23.5 °C and at 8.5 °C. Mixing ratios were measured using a 2B Technologies Model 400 NO Monitor. The low temperature was achieved by placing the calibrator in an ice chest, allowing it to cool and then powering the instrument on. The output was directed out of the ice chest and sampled by the NO monitor at room temperature. At startup, the





instrument showed that the photolysis chamber was at 8.5 °C. Results of measurements at the two
temperatures are summarized in Table 3. Data at the two temperatures agree very well within the standard
deviations of the measurements.

**Table 3**. Summary of effect of temperature on NO output concentration from a 2B Technologies
Model 408 NO Calibration Source.

| Target NO, ppb | Measured NO, ppb T = 23.5 °C | Measured NO, ppb T = 8.5 °C | Difference, ppb |
|---|---|---|---|
| 0 | 0.0 ± 2.8 | 0.0 ± 3.4 | 0.0 |
| 50 | 53.4 ± 2.9 | 55.9 ± 4.5 | 2.5 |
| 100 | 101.0 ± 2.7 | 101.1 ± 4.6 | 0.1 |
| 150 | 149.7 ± 2.8 | 151.8 ± 5.9 | 2.1 |
| 200 | 201.0 ± 1.9 | 198.5 ± 2.9 | -2.5 |
| **Average Difference:** | | | **0.5 ± 2.0** |

The average difference between measurements at the two temperatures is 0.5 ± 2.0 ppb; i.e., well
within the noise of the measurements. The average precision at 8.5 °C was ± 4.3 ppb compared to ± 2.6
ppb at 23.5 °C. Although a large fraction of this imprecision can be attributed to the 2B Technologies
Model 400 Nitric Oxide Monitor, it does appear that there is an increase in the measured standard deviations
at the lower temperature from the output of the calibrator. The increased power draw from heating the
photolysis chamber may affect the Hg lamp stability, causing this decrease in precision at lower
temperatures. We conclude that there is no significant dependence of the output concentration of the NO
calibrator on temperature in the range 8.5 to 23.5 °C; however, there is small loss of precision at lower
temperatures.

Lack of significant dependence of the NO calibrator of ambient pressure has been confirmed many
times by measuring the output NO mixing ratio of instruments calibrated in Boulder, Colorado (1.6 km
elevation, 844 mbar pressure) and shipped to other locations, typically at much lower elevations. In order
to extend the range of pressure testing to lower pressures, we measured the NO output in Boulder and at
Fritz Peak (2.7 km elevation, 745 mbar) near Rollinsville, Colorado. The NO calibrator and Model 400
NO Monitor were battery powered at the Fritz Peak location. Again, concentrations from 0 to 200 ppb
were measured at the two locations (see Fig. 7). As can be seen in the figure, within the precision of the
measurements, there is no discernible difference between the measurements at the two different altitudes.
Linear regressions of the measured NO values vs. NO setpoint (given in the inset panel of Fig. 7) indicate
a slightly lower slope (~ 3 %) at the higher elevation site. If there is a slight fall off in output concentration
at high altitudes, it could be explained by the lack of optical thickness within the $N_2O$ photolysis chamber

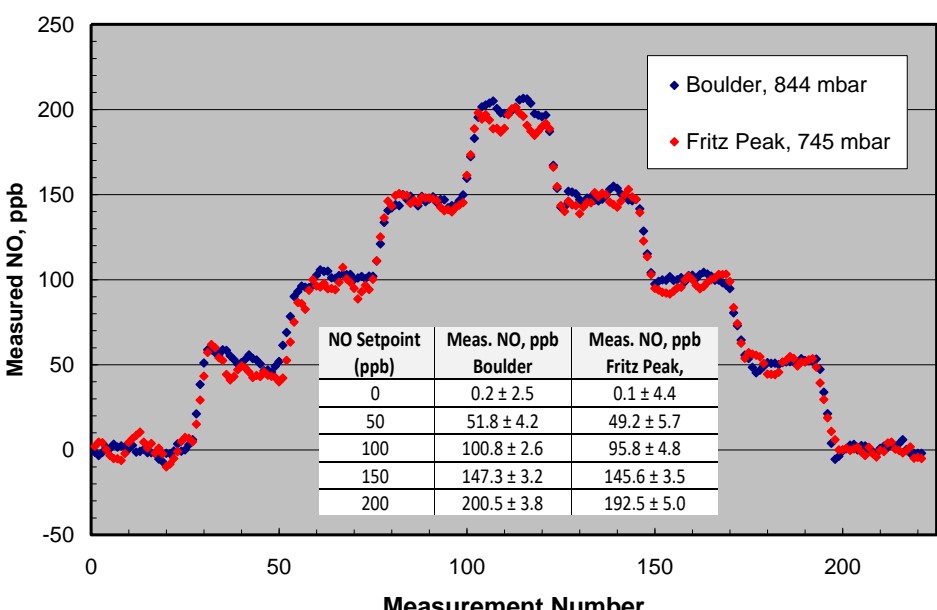

**Figure 7**. NO measured from the output of a Model 408 NO Calibration Source in Boulder, CO (elevation 1.6 km) and Fritz Peak, CO (elevation 2.7 km). The inset panel gives the averages and standard deviations (1σ) at the different NO setpoints of the calibrator. Measurements were made every 10 seconds.

due to the reduced pressure. The pressure and therefore molecular concentration are only slightly higher in the photolysis chamber than that of ambient air, so the fraction of 185-nm light absorbed decreases slightly as ambient pressure decreases.

450        Since pure $N_2O$ is the only gas passing through the photolysis chamber, ambient humidity should have no effect on the NO output rate. Humidity can only affect the overall NO output by affecting the output of the airflow mass flow meter causing small errors in the calculated dilution. Because water has a different heat capacity than air (~ 30 % larger), an airflow saturated with water vapor at 1 atm and 25 °C (saturation vapor pressure = 31.7 mbar, $H_2O$ mole fraction = 3.1 %) has a heat capacity that is about 0.9 %

higher than that of dry air. Since the mass flow rate measurement is proportional to heat capacity and the NO calibration source adjusts the lamp intensity to produce NO in proportion to the measured total mass flow rate, one could expect a small (~ 1 %) error in the output mixing ratio. This would likely be within the uncertainties of most analytical NO monitors. We tested the effect of humidity on the NO calibration source output by using a 2B Technologies Model 400 Nitric Oxide Monitor to measure step profiles of 0,

50, 100, 150, and 200 ppb at both 0 % and 100 % relative humidity (RH). The target humidities were generated by supplying the air inlet of the NO calibration source with zero air from a compressed gas

cylinder (0 % RH) and then humidifying that flow to ~ 100 % RH by passing it through a Nafion tube submerged in warm water. The 100 % RH experiment was run twice. Ambient temperature was 23.5 °C. For all experiments, the relative humidity was measured using a Cole Parmer Model 37951-00

Thermohygrometer inserted in line with the supply air flow. Plots of measured NO concentration vs. target concentration are shown in Fig. 8. The slopes of the regression lines were 0.968 at 0 % RH and 0.967 and 0.985 for two sequential calibrations made at 100 % RH. Within measurement error, there was no statistical difference between dry air and 100 % RH air, confirming our expectations that any humidity effect is within the statistical uncertainty of the analyzer.

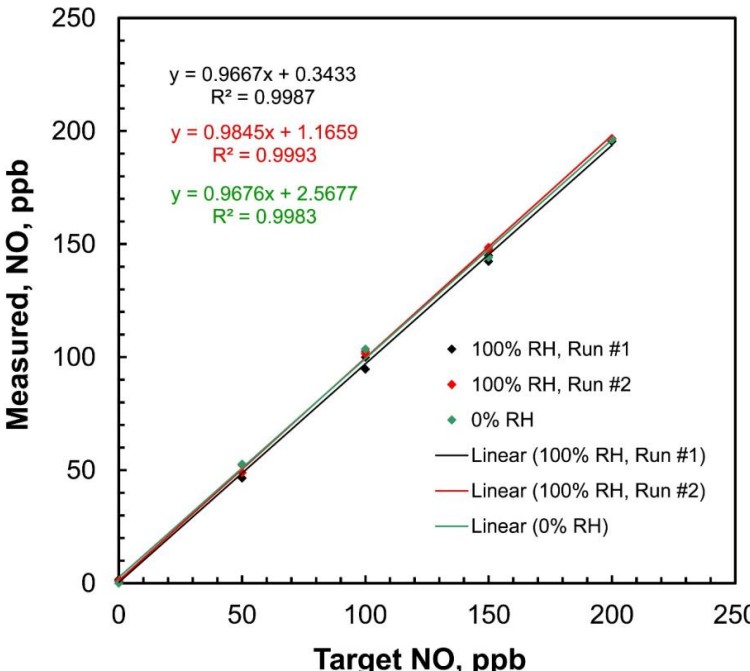

**Figure 8.** Linear regressions of measured NO concentrations vs. target concentrations of the 2B Technologies Model 408 NO Calibration Source with dilution air containing both 0 and 100 % relative humidity at 23.5 °C.

### 4.2    $NO_2/NO/O_3$ Calibration Source

470         The 2B Technologies Model 714 $NO_2/NO/O_3$ Calibration Source is a combination of a Model 306 Ozone Calibration Source and a Model 408 NO Calibration Source (Section 4.1) that allows for generation of calibrated mixing ratios of either $O_3$, NO or $NO_2$ (the latter via the GPT reaction 3). The NO source is identical to the one described in Section 3.1 and 4.1, except the $N_2O$ flow rate is typically lowered to around

40 $cm^3$ $min^{-1}$ compared to the earlier Model 408 NO Calibration Source. As noted in Section 3.1, the NO



output is not affected by the choice of $N_2O$ flow rate due to the optical opacity of the $N_2O$. The ozone calibration source in the Model 714 does differ from that described previously for our Model 306 (Birks et al., 2018b) in that only a small fraction of the airflow passes through the photolysis cell (only 50 cm$^3$ min$^{-1}$ as opposed to ~ 3 L min$^{-1}$). This increases the cell residence time from 0.06 s to ~ 3.6 s, and, consequently,

results in production of much higher ozone mixing ratios (up to 15 ppm) exiting the photolysis cell of the Model 714. However, control of the lamp intensity and volumetric flow rate (as described in Birks et al., 2018b) still allows for precise control of the output ozone mixing ratio that is independent of pressure and temperature. The longer residence time and higher mixing ratios in the $O_3$ photolysis cell do lead to complications due to water vapor that were not found in the individual $O_3$ calibration source (the Model

306). A solution to this potential problem will be discussed in the next section.

For the generation of $NO_2$, the outputs of the NO and $O_3$ photolysis cells are mixed and allowed to react in a ~6.5 cm$^3$ Teflon reaction volume. The total flow rate passing through this reactor is 90 cm$^3$ min$^{-1}$ (40 cm$^3$ min$^{-1}$ of NO/$N_2O$ and 50 cm$^3$ min$^{-1}$ of $O_3$/air) giving a reaction time of 4.3 s. Concentrations within the reaction zone can be calculated by knowing the measured output after dilution (i.e., the setpoint

concentration) and the ratio of the reaction zone and dilution flows. With a typical total flow (after dilution) of 2,700 cm$^3$ min$^{-1}$, initial reaction concentrations are a factor of 30 higher than the setpoint (or output) concentrations. For example, a final output concentration of 200 ppb of NO would give an initial concentration of 6 ppm of NO within the reaction zone.

***4.2.1    Water vapor effects and verification of the modified photolytic $O_3$ Calibration Source***

Section 4.1.2 showed that the effects of water vapor are very small (< 0.5 %) on the NO output from dry air up to air saturated with water vapor. Since the NO photolytic generator is unchanged in the $NO_2$/NO/$O_3$ calibrator, it also shows minimal effects due to changing humidity. However, the stand-alone ozone calibrator (Model 306) operates using rather different flow rates (and therefore, residence times) than

the ozone photolysis cell in the $NO_2$/NO/$O_3$ calibrator described here. Birks et al. (2018a) found that chemical loss of ozone due to OH and $HO_2$ radicals (generated either by water photolysis at 184.9 nm or by ozone photolysis and subsequent reaction of $O(^1D)$ with $H_2O$) was a negligible effect on the ozone output in the stand-alone ozone calibrator (the Model 306). The only effect of water vapor was the small dilution of the $O_2$ precursor by water vapor in the photolysis cell that results in a small of reduction of the ozone

generated (up to ~2 %). But the flow rate through the ozone photolysis cell in the $NO_2$/NO/$O_3$ calibrator is ~60 times slower than in the stand-alone Model 306 (50 cm$^3$ min$^{-1}$ compared to 3 L min$^{-1}$). Therefore, the longer residence time generates considerably higher concentrations of ozone, resulting in higher



concentrations of $HO_x$ (OH and $HO_2$) radicals when water vapor is present, which, in turn, can catalyze ozone depletion.

Modeling of the photolysis chemistry using the reaction kinetics model described in Birks et al. (2018b) suggested that at a relative humidity of 25 % (at 298 K), the ozone output in the $NO_2/NO/O_3$ calibrator would be reduced by 3.4 % when attempting to output 500 ppb (a loss of 17 ppb).  The ozone loss was also nonlinear – a smaller percent loss at lower $O_3$ setpoints.  This is due to the nonlinear nature of the $HO_x$ catalytic ozone destruction cycle that is driven by the high concentrations of ozone in the

photolysis chamber:

$$OH + O_3 \rightarrow HO_2 + O_2 \tag{11}$$

$$HO_2 + O_3 \rightarrow OH + 2\,O_2 \tag{12}$$

$$Net: 2\,O_3 \rightarrow 3\,O_2$$

    Experimental results showed that at low RH (RH = 6 – 10 %), the observed decreases in ozone

output in the $NO_2/NO/O_3$ calibrator relative to dry air (RH < 1 %) were on the order of 2-3 % for an output concentration of 500 ppb (i.e, 11 to 16 ppb).  This is in reasonable agreement with the ~1.5 % decrease predicted by the photochemical model.  However at a more typical relative humidity level of 25 %, observed ozone decreases were significantly greater than those predicted.   As mentioned above, predicted losses suggested a 3.4 % loss at 500 ppb; however, observations ranged from 6-12 % (34 to 60 ppb).  Therefore,

it appears there is even greater chemical loss than expected.  As a result of the very nonlinear nature of the chemistry when water vapor was present, it was necessary to dry the air prior to entering the ozone photolysis chamber of the $NO_2/NO/O_3$ calibrator.  An 80 $cm^3$ silica gel trap (United Filtration, IACH-38-150-80-SG) was added in line to reduce the RH to < 1 % in the $O_3$-precursor airflow (see Fig. 3).  A relative humidity/temperature sensor was also placed just before the ozone photolysis chamber to monitor the RH

and warn the user if the humidity rose to significant levels (RH > 2 %) such that ozone outputs could be impacted by more than 1 %.  At the typical flow rate of 50 $cm^3$ $min^{-1}$, this trap maintained the RH below 2% for more than 24 hours of continuous operation.  It should also be noted that once the air for the ozone photolysis has been dried, there is no significant amount of water vapor present in the GPT reaction zone, as the flow consists only of dry air/$O_3$ and dry $N_2O/NO$, thereby eliminating any possibility of water vapor

affecting the GPT chemistry.

    After the insertion of the dryer, a "6 × 6" verification was performed for this slightly modified ozone calibration source used in the $NO_2/NO/O_3$ calibrator.  Ozone concentrations were measured with a 2B Technologies Model 205 Ozone Monitor that had been recently calibrated relative to our primary ozone standard (Section 3.2.3).  Calibration plots and results of the linear regressions are shown in Fig. 9.  Both

the standard deviations in the slopes and intercepts are well within the U.S.-EPA transfer standard

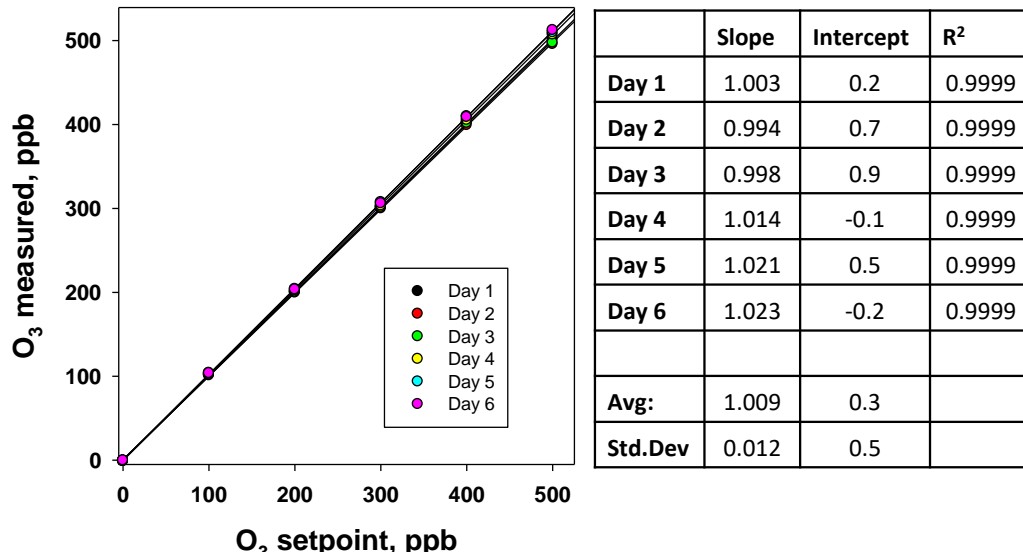

**Figure 9.** Plots of the $O_3$ measured vs. the $O_3$ setpoint in a Model 714 $NO_2/NO/O_3$ calibrator on 6 consecutive days. Regression slopes and intercepts are given in the panel on the right.

requirements ($\sigma_{slope} < 3.7$ %, $\sigma_{intercept} < 1.5$ ppb), thereby confirming that the adaptations made in the $O_3$ photolysis system for use in the $NO_2/NO/O_3$ calibrator do not adversely impact its use as an $O_3$ transfer standard.

### 4.2.2 Precision, accuracy and reproducibility of the NO₂ Calibration Source

Figure 10 shows mixing ratios of NO and $NO_2$ produced by a Model 714 $NO_2/NO/O_3$ calibration source (as measured by a Model 405 $NO_2/NO/NO_x$ Monitor) for an automated sequence of several $NO_2$ concentrations. The NO setpoint remained constant at 1000 ppb between 5 to 80 minutes during the sequence. Six different ozone concentrations (i.e., equal to the target $NO_2$ concentrations) were then generated (setpoints = 0, 80, 180, 280, 380, 480 ppb of ozone) each lasting 10 minutes (note, the use of 480

ppb instead of 500 ppb allows for visual clarity in the time series graph at the highest concentration). As seen in the figure, $NO_2$ increases as NO decreases due to its reaction with ozone. The time required to reach a new setpoint is typically < 45 seconds. The measured concentrations averaged over the last 5 minutes of each step are shown in the panel to the right of the figure along with observed precisions ($1\sigma$). Note that the precisions for steps 2-8 are nearly the same as those in steps 1 and 9 where no reagent gases were being

produced. This suggests that the observed precisions are limited by the measuring analyzer and that the

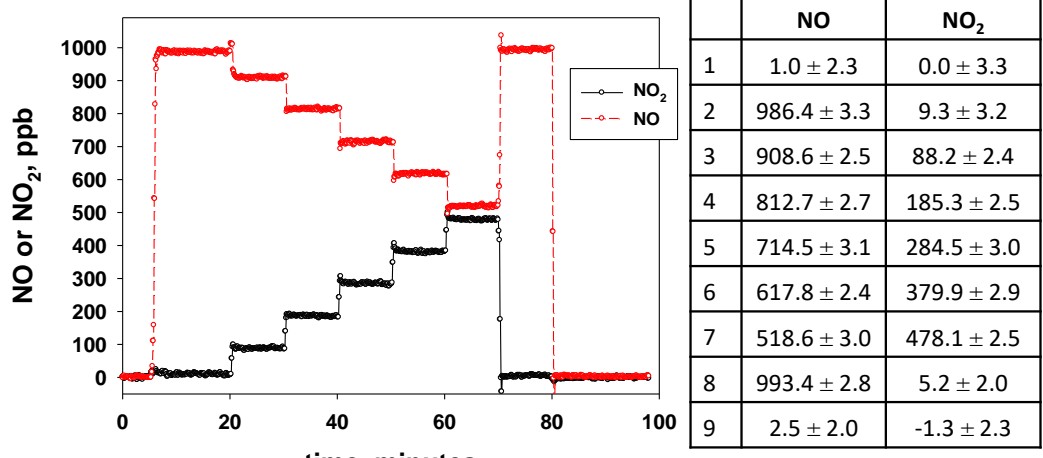

| | NO | NO$_2$ |
|---|---|---|
| 1 | 1.0 ± 2.3 | 0.0 ± 3.3 |
| 2 | 986.4 ± 3.3 | 9.3 ± 3.2 |
| 3 | 908.6 ± 2.5 | 88.2 ± 2.4 |
| 4 | 812.7 ± 2.7 | 185.3 ± 2.5 |
| 5 | 714.5 ± 3.1 | 284.5 ± 3.0 |
| 6 | 617.8 ± 2.4 | 379.9 ± 2.9 |
| 7 | 518.6 ± 3.0 | 478.1 ± 2.5 |
| 8 | 993.4 ± 2.8 | 5.2 ± 2.0 |
| 9 | 2.5 ± 2.0 | -1.3 ± 2.3 |

**Figure 10.** NO and NO$_2$ measured from the output of a Model 714 GPT calibrator. At time = 5 minutes, NO was set to 1000 ppb. At time = 20 minutes, ozone was added and setpoints were varied between 80 and 480 ppb in steps of 100 ppb (this allowed for visual clarity of both NO and NO$_2$ at the highest [O$_3$]). O$_3$ was changed every 10 minutes. Finally, data for [O$_3$] = 0 and [NO] = 0 were repeated for completeness. NO and NO$_2$ were measured using a calibrated 2B Tech Model 405.

actual precisions of the output NO and NO$_2$ concentrations from the calibrator are lower ($\leq \pm$ 2.8 ppb for NO, $\leq \pm$ 2.6 ppb for NO$_2$, the average precision from Fig. 10 panel).

  Also note the small amount of NO$_2$ produced by the NO photolysis source (9.3 ppb, ~ 0.9 % of the NO, see Fig. 10). As discussed in Section 4.2.3, this NO$_2$ is typically small ($\leq$ 2 % of the NO produced), and it is also constant over a given calibration with a set NO concentration. Therefore, a step where NO is present with no accompanying O$_3$ must be included to measure and subtract out this small amount of photolytically produced NO$_2$.

  Figure 11 shows the results of a 6 × 6 verification for NO$_2$ produced by the Model 714 NO$_2$/NO/O$_3$ calibrator along with the results from the linear regressions. As seen in the figure, plots were extremely linear (high R$^2$) with slopes near unity and reproducible from day to day. Relative standard deviations of the slope and intercepts were 0.4 % (compared with the required < 3.7 %) and 0.6 ppb (compared with the required < 1.5 ppb), respectively. Thus all three reactants (O$_3$, NO and NO$_2$) produced in the Model 714 NO$_2$/NO/O$_3$ calibrator pass the statistical tests established for a Level 4 ozone transfer standard.





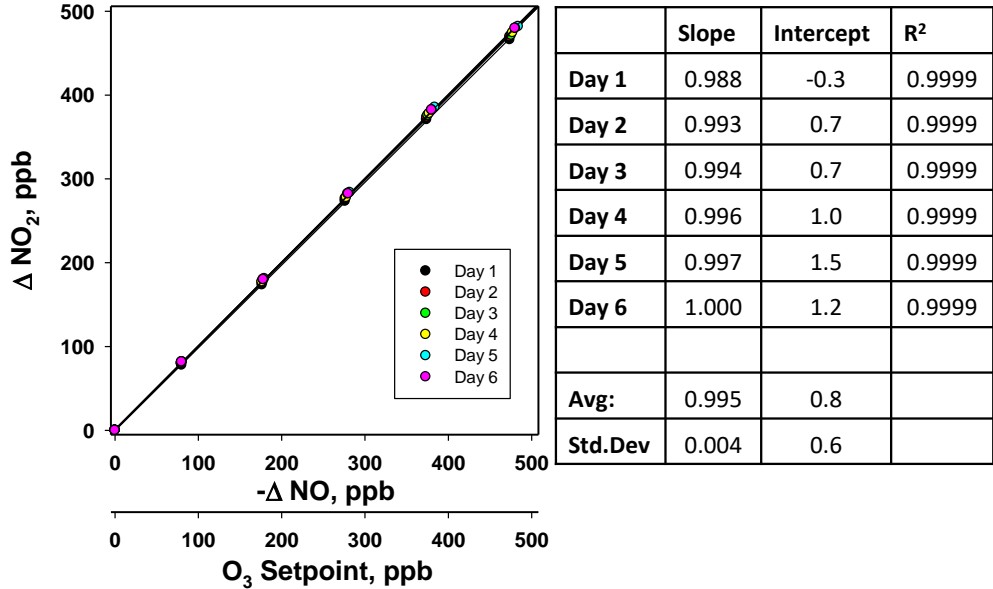

| | Slope | Intercept | $R^2$ |
|---|---|---|---|
| Day 1 | 0.988 | -0.3 | 0.9999 |
| Day 2 | 0.993 | 0.7 | 0.9999 |
| Day 3 | 0.994 | 0.7 | 0.9999 |
| Day 4 | 0.996 | 1.0 | 0.9999 |
| Day 5 | 0.997 | 1.5 | 0.9999 |
| Day 6 | 1.000 | 1.2 | 0.9999 |
| | | | |
| Avg: | 0.995 | 0.8 | |
| Std.Dev | 0.004 | 0.6 | |

**Figure 11.** Calibration curves for $NO_2$ made on 6 consecutive days. $\Delta NO_2 = NO_2 - NO_2(O_3 = 0)$ and $-\Delta NO = NO(O_3 = 0) - NO$ at each ozone setpoint. Slopes, intercepts and correlation coefficients are reported in the panel to the right.

*4.2.3 Completeness of reaction and effects of temperature and pressure*

Complete conversion of ozone to $NO_2$ is not critical if NO is measured as well (then $-\Delta[NO] = \Delta[NO_2]$ and the $NO_2$ signal can be calibrated relative to NO); however, the conversion efficiency is important in cases where the $NO_2$ produced must be compared to the ozone generated by the calibrated photolysis source. In a general sense, complete conversion also simplifies the chemical system and reduces

the chance for undesirable chemistry. Our modeling exercise (see Fig. 1) suggests that we achieve > 99 % reaction of the ozone for NO output concentrations greater than about 130 ppb (or ~ 4 ppm in the reactor) at 1 atm (1013 mbar) and 298 K and a reaction time of 4 seconds. We have shown that temperature and pressure do not affect the output mixing ratios from our photolytic sources of ozone and NO; however, these factors can impact the GPT chemistry occurring within the reaction zone. Reaction (3) has a rather

substantial activation energy ($E/R = 1,500$ $K^{-1}$, Burkholder et al., 2015) resulting in a smaller rate coefficient with decreasing temperature. Lower pressures have the effect of reducing the overall number density of the reactants, which is key to driving reaction (3) to completion. Although our photolytic sources ($O_3$ and NO) output constant *mixing ratios* with varying pressure, the *number density* (molec $cm^{-3}$) of ozone and nitric oxide *do vary* with overall pressure changes.





We tested this by measuring all three components ($O_3$, NO and $NO_2$) for a variety of setpoint concentrations from the calibrator ([NO] varied from 50 to 500 ppb, [$O_3$] from 24 to 240 ppb) while maintaining [NO] > 2 x [$O_3$].  Changes in $O_3$, NO and $NO_2$ were measured relative to when [$O_3$] = 0 (i.e., $\Delta NO = NO_{meas} - NO_{O3=0}$).  Table 4 shows the results from two experiments conducted at room temperature (25 °C, 298 K) and at 0 °C (273 K).  Both experiments were conducted at a total pressure of 830 mbar of

pressure (ambient pressure at our location in Boulder, Colorado, USA).   Note that total pressure closer to 1 atm (as would be typical) results in larger number densities, thus, driving a higher extent of reaction.  The temperature of the reaction zone was maintained by wrapping the reaction zone tubing in a flexible ice pack. The temperature was measured by attaching two thermocouples to the outside of the reaction zone tubing (one on each end).   The extent of the conversion of NO to $NO_2$ can be quantitated by looking at

either the loss of NO (-$\Delta NO/O_3$ generated) or the formation of $NO_2$ ($\Delta NO_2/O_3$ generated).  Here, the $O_3$ generated is the original setpoint of the ozone generator on the GPT calibrator.  Complete conversion results in -$\Delta NO = \Delta NO_2 = O_3$ setpoint (and, thus: -$\Delta NO/O_3$ setpoint = $\Delta NO_2/O_3$ setpoint = 1).  Figure 12 displays the results graphically by plotting the measured $\Delta NO/O_3$ and $\Delta NO_2/O_3$ (as percentages) vs. the initial NO setpoint of the GPT calibrator.  A second x-axis is included indicating the initial NO mixing ratio present

in the reaction zone (RZ), which can be used as reference to Fig. 1.  The only difference between the modeled profiles in Fig. 12 and those from Fig. 1 is that here the model was run under the experimentally observed temperatures and pressures.  Our measured results agree quite well with modeling of the chemistry.  Complete consumption (> 98 %) of the ozone was observed at NO setpoints above 200 ppb (~

**Table 4**.  $O_3$, NO and $NO_2$ measured at the output of a Model 714 for the NO and $O_3$ setpoints given in columns 1 and 2.  The pressure was 830 mbar.  All units are in ppb.

| NO setpt. | $O_3$ setpt. (=$NO_2$) | $\Delta NO^{[a]}$ | $\Delta NO_2^{[a]}$ | $\Delta O_3^{[a]}$ | Predicted[b] $O_3$ left |
|---|---|---|---|---|---|
| **298 K** | | | | | |
| **500** | **240** | $243.2 \pm 2.9$ | $240.7 \pm 3.2$ | $0.2 \pm 2.2$ | < 0.1 |
| **200** | **96** | $101.6 \pm 3.3$ | $97.8 \pm 3.4$ | $0.7 \pm 2.3$ | 0.2 |
| **100** | **48** | $46.8 \pm 3.0$ | $47.4 \pm 3.3$ | $1.6 \pm 3.1$ | 1.8 |
| **50** | **24** | $19.0 \pm 3.3$ | $20.0 \pm 3.0$ | $4.1 \pm 2.5$ | 3.8 |
| **273 K** | | | | | |
| **500** | **240** | $246.8 \pm 3.1$ | $242.5 \pm 2.4$ | $0.1 \pm 1.0$ | < 0.1 |
| **200** | **96** | $96.9 \pm 3.3$ | $96.2 \pm 2.7$ | $1.0 \pm 0.7$ | 1.3 |
| **100** | **48** | $46.9 \pm 3.0$ | $47.5 \pm 2.6$ | $3.1 \pm 0.7$ | 4.3 |
| **50** | **24** | $18.1 \pm 2.5$ | $18.1 \pm 2.5$ | $4.5 \pm 0.9$ | 6.2 |

[a]Concentration difference measured relative to when [$O_3$] = 0 (no ozone produced)
[b]From a model of $2^{nd}$ order kinetics of the reaction chamber chemistry at 830 mbar.



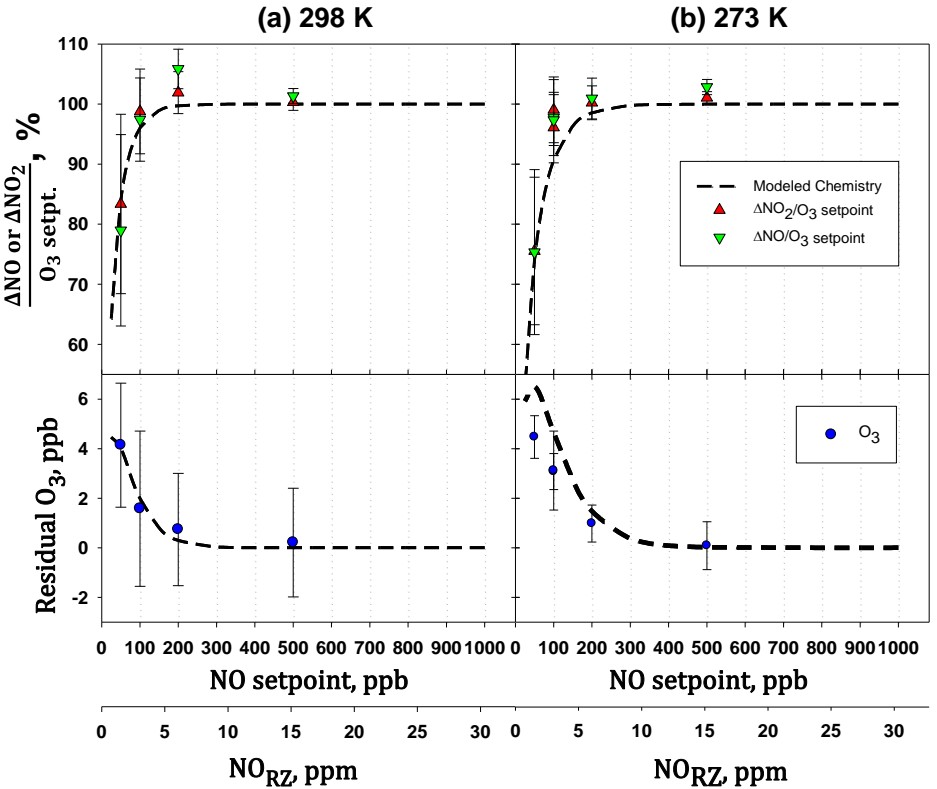

**Figure 12**. Plots of the amount of NO consumed ($\Delta NO/O_3$ setpt) and $NO_2$ formed ($\Delta NO_2/O_3$ setpt) vs. the NO setpoint. A second x-axis of the NO concentration in the reaction zone ($NO_{RZ}$) is also given as a reference to Fig. 1. Measurements were made with the reaction zone at (a) 298 K and (b) 273 K.

6 ppm in the RZ) at the lowest temperature (273 K). At room temperature, this level drops to near 100 ppb

(~ 3 ppm in RZ) within the errors of the measurements. Note that at the lowest initial NO and $O_3$ (NO setpoint = 50, $O_3$ setpoint = 25) where there is measurable residual ozone (~ 4-5 ppb), $\Delta NO_2$ is still equivalent to -$\Delta NO$ as expected from the stoichiometry of reaction (3). However, the percent error is considerably larger (~ 15 %) due to the smaller concentration changes and the measurement precision of the Model 405 $NO_2/NO/NO_x$ analyzer. It also bears reiterating at this point that it is critical to maintain at

least a factor of 2 greater NO to drive the $NO/O_3$ reaction to completion. Currently the operating firmware of the Model 714 does not allow $[NO]/[O_3] < 2$. Under these conditions, a general recommendation is that a NO setpoint of $\geq 200$ ppb ensures complete ozone consumption.





## 5. Conclusions

615       In the present study we have described two different portable calibration devices that can be used to calibrate air quality monitors. The first uses the photolysis of nitrous oxide to reproducibly generate known concentrations of NO (commercially available as the 2B Technologies Model 408 NO Calibration Source). The second combines this NO generator with a photolytic ozone generator (by photolysis of air) giving a single instrument capable of delivering calibrated mixing ratios of either NO, $NO_2$ or $O_3$ (the 2B

Technologies Model 714 $NO_2/NO/O_3$ Calibration Source). The chemistry underlying the generation of each reactant was discussed, and experimental results verified modeling predictions of the chemical systems involved.

      Since only a small amount of $N_2O$ is required, an 8- or 16-gram cartridge can be utilized as the source gas in either calibrator, thereby eliminating the need for larger (and more expensive) gas calibration

mixtures. This makes for the high degree of portability that is often necessary for the calibration of field-based analyzers that cannot easily be removed from service. Furthermore, we have shown that both the Model 408 and 714 produce calibrated mixing ratios that are independent of environmental variables such as temperature, pressure and humidity. This is also an advantage when operating in field situations where these variables are not controlled.

Both the NO and $NO_2/NO/O_3$ calibration systems are initially compared to NIST-traceable calibration standards (either NIST-SRM gas mixtures or NIST-traceable ozone generators/photometers) to establish the relationship between photolytic lamp intensity and output mixing ratios of NO, $NO_2$ and $O_3$. Once this is known, variation of the photolytic lamp intensities can reproducibly generate known concentrations of these reactants. The photolytic calibration systems were shown to deliver output mixing

ratios that were well within the guidelines required by the U.S.-EPA to serve as transfer calibration standards for these important pollutants both in terms of accuracy and precision. Therefore these calibrators can facilitate the calibration of analyzers at field locations where maintaining the high degree of accuracy and precision required by air quality compliance monitoring is challenging.

## 6. Data availability

Experimental data presented here are available upon request to the authors (johnb@twobtech.com).

## 7. Competing interests

      John Birks, Andrew Turnipseed, Peter Andersen, Craig Williford, Stanley Strunk, Brian Carpenter and Christine Ennis are employed by 2B Technologies, the manufacturer of the Model 408 NO Calibration

Source™ and the Model 714 $NO_2/NO/O_3$ Calibration Source™ described in this paper.



## 8.    Acknowledgments

Aerocrine AB of Stockholm, Sweden provided funding for development of the Model 408 Nitric Oxide Calibration Source.

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
