# Peer review of "Portable Calibrator for NO Based on Photolysis of N2O and a Combined NO2/NO/O3 Source for Field Calibrations of Air Pollution Monitors"

_Atmospheric Measurement Techniques, 2019_

## Referee Comment (RC1) · Anonymous Referee #1 · 21 Nov 2019

The manuscript describes a portable calibration system/transfer standard for NOx monitors based upon photolysis of N2O from a small compressed gas source. A GPT system allows for calibration of NO2 conversion efficiency as well as NO sensitivity as is often required in the most common NOx air monitors employing NO chemiluminescence. Ozone monitors may also be calibrated with the same unit. The calibration system appears very well characterized and robust also the manuscript is clearly written, well presented, detailed, referenced etc. I recommend publication in AMT once a few relatively minor issues have been addressed.

General Comments

[Figure]

Stylistically there are lots of references to 'our' or 'we' in the manuscript which I would rather are depersonalized throughout.

In a similar vein, the manuscript strays into becoming an advertisement of seemingly the entire 2B Technologies range of products. This is to be expected and is possibly unavoidable though it seems unnecessary to describe both the Model 408 and also the Model 306 (already described in Birks et al., 2018b) separately. The obvious competing interest of the authors is rightfully declared however.

Specific comments

P2 L60 – "...a strong tendency for the concentration of NO in the cylinder to decline with time..." – please provide a reference e.g. Robertson et al 1977 (https://doi.org/10.1080/00022470.1977.10470491) or similar of the authors choosing. This is oft-claimed without reference.

P25 L571 "Complete conversion of ozone to NO2 is not critical if NO is measured as well..." True, except if calibrating a photolytic NO2 converter whose conversion is a function of J and concentration of oxidants of NO. However, below it is shown that O3 is negligible.

Technical Corrections

P8 L226 "photolytic NO converters" should be "photolytic NO2 converters" I think

---

## Referee Comment (RC2) · Anonymous Referee #2 · 11 Dec 2019

**Review of Birks et al., AMTD, https://doi.org/10.5194/amt-2019-399**

This manuscript reports specifications and characterization tests for portable calibrators dedicated to the generation of NO and $NO_2/NO/O_3$ transfer standards. A special attention was put on assessing their performance in terms of stability, reproducibility, precision and accuracy. An originality of this work is the use of $N_2O$ photolysis to generate a known concentration of NO, which when combined to a previously described $O_3$ generator, allows the generation of a known concentration of $NO_2$ using the gas phase titration technique. The authors showed that these calibrators meet the US-EPA requirements for transfer calibration standards (based on the requirements for a level 4 ozone transfer standard).

This manuscript is well structured and well written but some clarifications need to be made about the different analyzers that were used to perform all the reported experiments. Different models of monitors were used and when the same model was used in different experiments, it is not clear whether it was the same monitor. In addition, the authors often indicate that the measurements were made using a "recently calibrated XXX monitor". How was the monitor calibrated? The authors should clarify these two points.

The reviewer recommend publication in AMT after the authors address the other following comments:

**Major comment:** Since these apparatus will be used for the calibration of NOx monitors based on chemiluminescence, the authors should address whether the use of a large concentration of $N_2O$ could lead to the quenching of the chemiluminescence.

**Minor comments:**

P8 L233: Please indicate what types of $O_3$ and $NO_x$ scrubbers were used

P9 L249-250: The authors indicate that it is important to maintain a constant ratio of lamp emission between 184.9 and 253.7 nm. How is this ratio changing with the lamp aging? In addition, it is mentioned that NO is varied by adjusting the lamp emission. How is it done? If the current or the voltage applied to the lamp is varied, the authors should comment on the impact on the 184.9nm/253.7nm emission ratio?

P16 Fig.6: The repeatability for NO generation tested over 6 days is excellent. Based on their practical experience, could the authors comment on the repeatability over a longer period? Weeks? Months?

**Technical corrections:**

P6 L172-173: "O" should read "O($^3$P)"

P5 L173: M is missing on the right hand side of the equation

P8 L226: "photolytic NO converter" should read "photolytic $NO_2$ converter"

P18: L436: "the NO calibrator of ambient pressure" should read "the NO calibrator on ambient pressure"

P 21 L476-494: This section should be moved in 3.2 when the calibration sources are first described.

P22 L 539: Section 3.2.3 does not exist. Should it read "section 3.3"?

P24 Fig. 10: Please add the $O_3$ setpoints in the table.

P26 L588: Please replace "$NO_{meas}$" by "$NO_{meas,O3>0}$" and "$NO_{O3=0}$" by "$NO_{meas,O3=0}$"

---

## Author Response (AR1)

AMT-2019-399: Portable Calibrator for NO Based on Photolysis of N$_2$O and a Combined NO$_2$/NO/O$_3$ Source for Field Calibrations of Air Pollution Monitors

**Reply to the review by Anonymous Referee #1:**

We appreciate the time and efforts of Referee #1. We would like to thank them for contributing their thoughtful comments. Their comments are listed below in **Bold** font followed by our responses in blue text. New text added is given in italics (quoted) along with page and line numbers within the new revised manuscript.

**Reviewer #1.**
**The manuscript describes a portable calibration system/transfer standard for NOx monitors based upon photolysis of N2O from a small compressed gas source. A GPT system allows for calibration of NO2 conversion efficiency as well as NO sensitivity as is often required in the most common NOx air monitors employing NO chemiluminescence. Ozone monitors may also be calibrated with the same unit. The calibration system appears very well characterized and robust also the manuscript is clearly written, well presented, detailed, referenced etc. I recommend publication in AMT once a few relatively minor issues have been addressed.**

**General Comments**
**Stylistically there are lots of references to 'our' or 'we' in the manuscript which I would rather are depersonalized throughout. In a similar vein, the manuscript strays into becoming an advertisement of seemingly the entire 2B Technologies range of products. This is to be expected and is possibly unavoidable though it seems unnecessary to describe both the Model 408 and also the Model 306 (already described in Birks et al., 2018b) separately. The obvious competing interest of the authors is rightfully declared however.**

After reviewing our writing style, we agree that the "our" and "we" were overused and often unnecessary. We have removed the majority of these – the remaining ones typically describe particular points that "we" (as the authors) are trying to convey either (i.e., conclusions or major points). We do feel it necessary to describe the Model 408 (the NO photolytic source) separate from the GPT calibrator since we have had more testing and experience with that instrument. We limited our discussion of the Model 306 to just describe the important points outlined in Birks et al., 2018b and the changes that were necessary for use in the GPT calibrator described here.

**Specific comments**
**P2 L60 – ". . .a strong tendency for the concentration of NO in the cylinder to decline with time. . ."**
**– please provide a reference e.g. Robertson et al 1977**
**(https://doi.org/10.1080/00022470.1977.10470491) or similar of the authors choosing.**
**This is oft-claimed without reference.**
We have included this reference as suggested. We thank Reviewer #1 for noticing this.

P25 L571 "Complete conversion of ozone to NO2 is not critical if NO is measured as well. . ." True, except if calibrating a photolytic NO2 converter whose conversion is a function of J and concentration of oxidants of NO. However, below it is shown that O3 is negligible.
We agree with Reviewer #1 and this is an important reason to limit the amount of ozone exiting the calibrator. At this point (page 26, line 607), we have added the text (and reference):
*"It is also important to limit the amount of ozone exiting the calibrator in the case of NO$_x$ analyzers that use a photolytic NO$_2$ converter as the NO$_2$ conversion efficiency of these converters is known to depend upon ozone concentration (e.g., see Pätz et al., 2000). "*

**Technical Corrections**
**P8 L226 "photolytic NO converters" should be "photolytic NO2 converters" I think**
Corrected as suggested.

AMT-2019-399: Portable Calibrator for NO Based on Photolysis of $N_2O$ and a Combined $NO_2/NO/O_3$ Source for Field Calibrations of Air Pollution Monitors

**Reply to the review by Anonymous Referee #2:**

We appreciate the time and efforts of Referee #2. We would like to thank them for contributing their thoughtful comments. Their comments are listed below in **Bold** font followed by our responses in blue text. New text added is given in italics (quoted) along with page and line numbers within the new revised manuscript.

**Reviewer #2:**
**This manuscript reports specifications and characterization tests for portable calibrators dedicated to the generation of NO and NO2/NO/O3 transfer standards. A special attention was put on assessing their performance in terms of stability, reproducibility, precision and accuracy. An originality of this work is the use of N2O photolysis to generate a known concentration of NO, which when combined to a previously described O3 generator, allows the generation of a known concentration of NO2 using the gas phase titration technique. The authors showed that these calibrators meet the US-EPA requirements for transfer calibration standards (based on the requirements for a level 4 ozone transfer standard).**
**This manuscript is well structured and well written but some clarifications need to be made about the different analyzers that were used to perform all the reported experiments. Different models of monitors were used and when the same model was used in different experiments, it is not clear whether it was the same monitor. In addition, the authors often indicate that the measurements were made using a "recently calibrated XXX monitor". How was the monitor calibrated? The authors should clarify these two points.**
The different monitors for ozone and $NO_x$ (NO and $NO_2$) used are described in Section 3.3 (page 12, lines 338-347). This paragraph also describes how these analyzers were calibrated and how these calibrations were traceable to NIST standards. Different actual analyzers were used throughout the various experiments; however, each analyzer was calibrated as described just prior to use.

**The reviewer recommends publication in AMT after the authors address the other following comments:**

**Major comment:**
**Since these apparatus will be used for the calibration of NOx monitors based on chemiluminescence, the authors should address whether the use of a large concentration of N2O could lead to the quenching of the chemiluminescence.**
This is an excellent point that we had not considered. Clyne et al., (1964) measured the relative quenching rate constants (relative to $O_2 = 1$) of the $NO + O_3$ chemiluminescent system and report: $N_2 = 0.9$ and $N_2O = 2.6$. (i.e., $N_2O$ quenches 2.6 times more effectively than $O_2$ and 2.9 times more than $N_2$). Therefore the total relative quenching in pure air is:
$$kq(air) = (\%N_2)*0.9 + (\%O_2)*1 = 0.9122 \text{ (where } N_2 = 0.7808 \text{ and } O_2 = 0.2095).$$

Addition of 1.5% $N_2O$ (typical of our calibrators) yields an effective quenching constant of
$$kq(eff) = (\%N_2O)*2.6 + (1 - \%N_2O)*0.9122 = 0.9375.$$

The ratio: kq(air)/kq(eff) = 0.973 suggests that the CL signal should be reduced by 2.7% (= 1 – 0.973). In practice, it is usually slightly less than this as CL-analyzers typically dilute the sample flow by addition of an $O_3$/air reagent (typically 10-25% of the total flow).

We also note here that the photolytic NO generator must initially be calibrated relative to a standard reference method (described in Section 3.1) using some type of NO analyzer. If that analyzer is a CL-analyzer, then the additional quenching by $N_2O$ will be incorporated into the calibration curve of lamp intensity vs. NO output. We have added the following text at the end of Section 2.1 (page 6, lines 166-176) describing this:

*"A final issue pertains specifically to CL NO analyzers where the presence of $N_2O$ can lead to collisional quenching of the chemiluminescence signal. Clyne et al. (1964) report that $N_2O$ quenches the $NO/O_3$ chemiluminescence 2.6 and 2.9 times more efficiently than $O_2$ and $N_2$, respectively. Therefore, a mixture of 1.5% $N_2O$ in air (typical conditions – see Section 3) would be expected to reduce the observed chemiluminescent signal by ~ 2.7% relative to pure air. In practice, this is typically slightly less (2.2-2.4%) as the sample flow in a CL-analyzer is diluted by addition of a reagent $O_3$/air flow (10-25% of the total flow). This correction term can be explicitly calculated from the measured flows in the photolytic NO calibration source and the flow rates in the CL-analyzers (as in the example given here) or it can be eliminated depending upon the analyzer used during the initial calibration of the relationship between lamp intensity vs. NO output (see Section 3.1). If a CL-analyzer is used to determine this relationship, then quenching by $N_2O$ is intrinsically included in the calibration of the photolytic source."*

**Minor comments:**
**P8 L233: Please indicate what types of O3 and NOx scrubbers were used**
We have noted that the scrubbers were "*…a mixture of Carulite® and activated carbon…*)

**P9 L249-250: The authors indicate that it is important to maintain a constant ratio of lamp emission between 184.9 and 253.7 nm. How is this ratio changing with the lamp aging? In addition, it is mentioned that NO is varied by adjusting the lamp emission. How is it done? If the current or the voltage applied to the lamp is varied, the authors should comment on the impact on the 184.9nm/253.7nm emission ratio?**
The lamp emission is varied by changing the pulse-width modulated duty cycle to the Hg lamp. We have added the sentence at pg10, line 265:
"*NO output is varied by changing the pulse-width modulated duty cycle to the Hg lamp while monitoring its intensity with a photodiode.*"

Our experience has shown that the ratio of the 184.9nm/253.7 nm emission is mainly dependent on the lamp temperature. Therefore, we control the temperature at 40°C in both our NO and $O_3$ photolysis cells. As the duty cycle to the lamp is varied (which can cause slight heating or cooling), the temperature control compensates to maintain a constant temperature – thus a constant ratio of 184.9/253.7 nm. The observation that the NO or $O_3$ output is strictly linear as one varies the duty cycle strongly suggests that the 184.9/253.7 nm does not change significantly. To clarify this, we have added the sentence at pg 10, line 272:
"*…Even as the duty cycle is varied, the temperature regulation maintains a constant lamp temperature, thus ensuring a stable 184.9/253.7 nm output.*"
Lamp aging is addressed in the following comment concerning long-term repeatability.

**P16 Fig.6: The repeatability for NO generation tested over 6 days is excellent. Based on their practical experience, could the authors comment on the repeatability over a longer period? Weeks? Months?**

We recommend that the photolytic NO generator should be re-calibrated at least annually (however, note that a Level 4 EPA transfer standard which is a generator-only is required to be re-calibrated quarterly - see US-EPA, 2013 in manuscript). For our ozone generators (for which we have more experience), we have observed that their calibration typically changes by $< 2\text{-}3\%$ over a yearly period. We would expect similar performance for the NO generator since it uses the same lamp and electronics. This exact cause of the small observed calibration drift is not explicitly known, but small changes in the $184.9/253.7$ nm ratio of the lamp could play a role.

**Technical corrections:**
**P6 L172-173: "O" should read "O($^3$P)"**
Corrected as suggested.

**P5 L173: M is missing on the right hand side of the equation**
Corrected as suggested.

**P8 L226: "photolytic NO converter" should read "photolytic NO2 converter"**
Corrected as suggested.

**P18: L436: "the NO calibrator of ambient pressure" should read "the NO calibrator on ambient pressure"**
Changed to: "…*dependence of the NO calibrator with ambient pressure*…"

**P 21 L476-494: This section should be moved in 3.2 when the calibration sources are first described.**
We agree with Reviewer #2 that this section is repetitive and have incorporated this text into Sections 2.2 (pg 7, lines 200-208) and Section 3.2 (pg. 11, lines 303-307, pg. 12, lines 316-318) while removing any redundancies in the information presented.

**P22 L 539: Section 3.2.3 does not exist. Should it read "section 3.3"?**
Corrected to "Section 3.3" as suggested.

**P24 Fig. 10: Please add the O3 setpoints in the table.**
Corrected as suggested.

**P26 L588: Please replace "NO$_{meas}$" by "NO$_{meas,O3>0}$" and "NO$_{O3=0}$" by "NO$_{meas,O3=0}$**
Corrected as suggested.

[revised manuscript text omitted]
 of N$_2$O at 193 nm and O($^3$P) + N$_2$O product channel in the reaction of O($^1$D) + N$_2$O, *J. Phys. Chem. A,* 108, 2451-2456, DOI: 10.1021/jp037034o, 2004.

Paldus, B.A. and Kachanov, A.A.: Spectroscopic Techniques: Cavity-Enhanced Methods, In *Handbook of*
765    *Atomic, Molecular, and Optical Physics, Part C: Molecules*, edited by G.W.F. Drake, Springer, Berlin, 633-640, 2006.

Pätz, H-W., Corsmeier, U., Glaser, K., Vogt, U., Kalthoff, N., Klemp, D., Kolahgar, B., Lerner, A., Neininger, B., Schmitz, T., Schultz, M.G., Slemr, J., and Volz-Thomas, A.: Measurements of trace gases
770    and photolysis frequencies during SLOPE96 and coarse estimate of the local OH concentration from HNO3 formation, *J. Geophys. Res*. 105, 1563-1583, 2000.

Ridley, B.A. and Howlett, L.C.: An instrument for nitric oxide measurements in the stratosphere, *Rev. Sci. Instrum.*, 45, 742-746, 1974.
775

Robertson, D.J., Groth, R.H., Gardner, D.G. and Glastris, G.: Stability and analyses of nitric oxide in nitrogen, *J. Air Pollut. Control Assoc.*, 27, 779-780, DOI: 10.1080/00022470.1977.10470491, 1977.

Steffenson, D.M. and Stedman, D.H.: Optimization of the operating parameters of chemiluminescent nitric
780    oxide detectors, *Anal. Chem.*, 46, 1704-1709, 1974.

U.S.-EPA: Reference Method for the Determination of Nitrogen Dioxide in the Atmosphere (Chemiluminescence), Quality Assurance Guidance Document 2.3, 58 pp., 2002.

785    U.S.-EPA: Transfer Standards for Calibration of Air Monitoring Analyzers for Ozone, Technical Assistance Document, Ozone Transfer Standard Guidance Document 10/2013, Publication No. EPA-454/B-13-004, Office of Air Quality Planning and Standards, Air Quality Assessment Division, Research Triangle Park, North Carolina, 68 pp., October 2013. https://www3.epa.gov/ttn/amtic/files/ambient/qaqc/OzoneTransferStandardGuidance.pdf, Visited October
790    10, 2019.

U.S.-EPA: Measurement Principle and Calibration Procedure for the Measurement of Nitrogen Dioxide in the Atmosphere (Gas Phase Chemiluminescence). U.S. Environmental Protection Agency. 40 CFR, Part 50, Appendix F, as amended Jan. 20, 1983.
795

U.S.-EPA: NO$_2$ cylinder guidance for state-local agencies and gas producers, https://www.epa.gov/air-research/no2-cylinder-guidance-state-local-agencies-and-gas-producers, and included pdf links, visited Sept. 5, 2019.

800    Vranckx, S., Peeters, J. and Carl, S.A.: Absolute rate constant and O($^3$P) yield for the O($^1$D) + N$_2$O reaction in the temperature range 227 K to 719 K, *Atmos. Chem. Phys.*, 8, 6261-6272, 2008.

Yoshino, K., Esmond, J.R., Cheung, A.S.C., Freeman, D.E. and Parkinson, W.H.: High resolution absorption cross sections in the transmission window region of the Schumann-Runge bands and Herzberg continuum of $O_2$, *Planet. Space Sci.*, 40, 185-192, 1992.

805